# Role of Echocardiography in the Diagnosis and Interventional Management of Atrial Septal Defects

**DOI:** 10.3390/diagnostics12061494

**Published:** 2022-06-18

**Authors:** P. Syamasundar Rao

**Affiliations:** Children’s Heart Institute, University of Texas-Houston McGovern Medical School, Children’s Memorial Hermann Hospital, UTPB Suite # 425, Houston, TX 77030, USA; p.syamasundar.rao@uth.tmc.edu or srao.patnana@yahoo.com; Tel.: +1-713-500-5738; Fax: +1-713-500-5751

**Keywords:** echocardiography, Doppler, atrial septal defect, device occlusion, Amplatzer Septal Occluder, Gore HELEX devices, Cribriform device, follow-up results, residual shunts

## Abstract

This review centers on the usefulness of echo-Doppler studies in the diagnosis of ostium secundum atrial septal defects (ASDs) and in their management, both in children and adults. Transthoracic echocardiography can easily identify the secundum ASDs and also differentiate secundum ASDs from other kinds of ASDs, such as ostium primum ASD, sinus venosus ASD and coronary sinus ASD, as well as patent foramen ovale. Preliminary selection of patients for device occlusion can be made by transthoracic echocardiograms while final selection is based on transesophageal (TEE) or intracardiac (ICE) echocardiographic studies with optional balloon sizing of ASDs. TEE and ICE are extremely valuable in guiding device implantation and in evaluating the position of the device following its implantation. Echo-Doppler evaluation during follow-up is also useful in documenting improvements in ventricular size and function, in assessing the device position, in detecting residual shunts, and in identifying rare device-related complications. Examples of echo images under each section are presented. The reasons why echo-Doppler is very valuable in diagnosing and managing ASDs are extensively discussed.

## 1. Introduction

Several varieties of atrial septal defects (ASDs) exist and these are ostium secundum, ostium primum, sinus venosus, and coronary sinus ASDs. In this review, only ostium secundum ASDs, hereafter referred to as ASDs, will be discussed. The prevalence of these defects is 9 to 13% of all congenital cardiac defects [1,2,3,4]. The ASDs are secondary to the absence of cardiac tissue in the region of fossa ovalis. The ASDs vary from small to medium and large in size. These defects are usually single defects, although, on occasion, several defects or an atrial septum with multiple fenestrations are also observed. Left-to-right shunting through the ASD results in dilation of the right heart structures, such as right atrium (RA), right ventricle (RV), and pulmonary arteries (PAs). Pulmonary hypertensive disease is typically seen in adults [5]. Patent foramen ovale is seen in 25% to 30% of the normal population and is generally thought to be a variation of the norm and will not be discussed in this review. The pathologic, patho-physiologic, clinical, roentogenographic, electrocardiographic, and cineangiographic features of ASDs were examined in the author’s prior publications [1,2,3] and will not be reviewed in this paper. This review discusses the role of echo-Doppler investigation in the diagnosis of ASDs, as well as their management, both in pediatric and adult populations.

## 2. Diagnosis

Transthoracic echocardiographic (TTE) examination is very useful in making the diagnosis of ASD in babies and children, as well as teenagers and adult subjects with thin build. In adolescents and adults with poor precordial and subcostal echo windows, transesophageal or intracardiac echocardiography is necessary to demonstrate all features of ASD. The pathophysiologic features of ASDs, namely dilatation of the RA, RV, and PAs (Figure 1), along with paradoxical (or flat) ventricular septal motion (Figure 2), can easily be demonstrated on echocardiogram.

The ASD can be clearly visualized by two-dimensional (2D) echocardiograms (Figure 3A and Figure 4A). In normal subjects, the central section of the atrial septum is thin and may not be clearly seen in 2D; these “septal drop-outs” may be falsely interpreted as ASDs, particularly in adolescents and adults. Therefore, subcostal views (which allow placement of the atrial septum perpendicular to the ultrasound beam) should be scrutinized for evidence of ASD. Furthermore, it is important to demonstrate flow across this site by color flow mapping (Figure 3B and Figure 4B) and/or by pulsed Doppler (Figure 5). In adolescents and adults, TEE or ICE may become necessary to formulate a clear-cut diagnosis of ASD.

As mentioned above, some patients have fenestrated atrial septal defects; such fenestrated defects can also be demonstrated by transthoracic echocardiographic studies (Figure 6).

A composite of images from subcostal views of an atrial defect with left-to-right shunting is illustrated in Figure 7. 

Two-dimensional and color flow imaging in several views should also be carried out to identify the entrance of each pulmonary vein into the left atrium (LA) and to exclude partial anomalous pulmonary venous return. Peak Doppler velocity of tricuspid insufficiency jet and end-diastolic velocity of pulmonary insufficiency jet are useful in calculating the pressures in the PA with the usage of a modified Bernoulli equation: PA Pressure = 4V^2^ + 5 mmHg

PA, pulmonary artery; V, Doppler flow velocity.

Determination of PA pressures is important, particularly because the closure of ASD is not indicated in subjects with pulmonary hypertension, specifically in patients with pulmonary vascular obstructive disease.

Natural reduction in the diameter of the ASD or complete closure is well documented in the literature [8,9], and the echo studies are useful in demonstrating such spontaneous closures. Restrictive ASDs with small size (a few mm) with increased Doppler velocity, reflecting the pressure difference between LA and RA, are also easily identified by the transthoracic echo studies. These patients do not exhibit right-heart volume overloading, described above.

TTE studies are also useful in distinguishing ostium secundum ASD (Figure 3, Figure 4 and Figure 7) from ostium primum (Figure 8), sinus venosus (Figure 9), and coronary sinus ASDs and patent foramen ovale (Figure 10).

## 3. ASD Closure Methods

Surgical closure was the conventional and favored treatment for ASDs [10,11,12,13] until the wider application of the transcatheter techniques described by King, Rashkind, and their associates [14,15,16,17,18] came into vogue. Historical aspects of ASD device closure are reviewed elsewhere [19,20,21,22] for the interested reader. Based on these and subsequent reviews, there are only few ASD occluding devices approved by the Food and Drug Administration (FDA) of the USA and these are: Amplatzer Septal Occluder (St. Jude Medical, Inc., St. Paul, MN, USA—Abbott), Gore HELEX^®^ device (W.L. Gore, Flagstaff, AZ, USA), Amplatzer Cribriform device (St. Jude Medical, Inc.), and GORE^®^ CARDIOFORM-ASD-Occluder (W.L. Gore).

## 4. Patient Selection for Device Occlusion

The current guidelines for ASD closure were reviewed by respective Committees of the American College of Cardiology, American Heart Association, and European Society of Cardiology [23,24]. Once the indications for ASD closure, i.e., volume overloading of the right heart (enlargement of RA and RV with paradoxical (or flat) ventricular septal motion) by echo (Figure 1 and Figure 2), are fulfilled, further evaluation for suitability for transcatheter occlusion should be made. Multiple 2D echo projections, mainly the subxiphoid and four-chamber views, are examined to measure the largest size of the ASD, the sizes of the rims of the defect, and the atrial septal length. Patients with adequate septal rims (Figure 3, Figure 4, Figure 7 and Figure 11) are considered suitable for closure, while those with no adequate inferior or superior rims, or both (Figure 12), are not suited for occlusion with a device. The length of the atrial septum should also accommodate the size of the device selected. 

We have, in the past, attempted to develop objective echocardiographic features that might forecast the successful device closure of an ASD [27]. In this study, the echo information secured was the size of the ASD (by 2D and color jet width), the length of the atrial septum (LAS), the size of the inferior and superior margins in two (apical and subcostal) projections (Figure 13), and several derived ratios (see Table 1 and Table 2 from our publication [27]). Multivariate logistic regression examination was performed and contingency tables were also developed. This analysis identified a size of ASD less than 15 mm, an ASD/LAS ratio less than 0.35, and a fraction of the superior rim to the ASD greater than 0.75 as factors likely to predict effective device implantation across the atrial defect [27]. The above study was undertaken for the implantation of a buttoned device and it is not certain if such criteria are equally applicable for Amplatzer Septal Occluder (ASO) and Gore HELEX device occlusions. To the author’s knowledge, no such studies for these devices were undertaken.

At the present time, the ultimate decision to insert the device is made after detailed TEE or ICE evaluation of the diameter and shape of the ASD and rims of the ASD and balloon sizing of the ASD (as will be detailed in the subsequent sections) in the Cardiac Catheterization Laboratory.

## 5. Three-Dimensional Echocardiogram

Three-dimensional (3D) echocardiographic studies were initially developed in the early 1990s [28,29], as reviewed elsewhere [30]. Three-dimensional echo appears to have an advantage in accurately measuring the ASD size in complex-shaped ASDs [31,32]. For circular-shaped ASDs, the measurements of ASD size are similar, both by 2D and 3D echocardiography, while in oval-shaped and bizarre-shaped ASDs, the 3D measurements were larger than those secured by 2D echo. Septal rims are more clearly characterized by 3D than by 2D echo [31,32]. Because of the usefulness and accuracy, some centers have routinely used 3D TEE during device closure of ASDs [33]. Other studies indicated that 3D is useful in ASD occlusion without the need for balloon sizing [34,35]. It may be concluded that 3D echo is a very promising and new echocardiographic technique, allowing appreciation of complex spatial relationships. The disadvantages of 3D echocardiography are low temporal and spatial resolution and the need for offline processing. However, the newest software allows one to demonstrate 3D images during the initial recording of the images. Nonetheless, the high device implantation rates (88 to 94%) [36,37] without the use of 3D echo lead the author to recommend that regular use of 3D is not needed; however, it might be used in patients with bizarre-shaped ASDs and those defects that have deficient septal rims [38], either on TTE or TEE studies.

## 6. TEE and ICE

The standard practice at this time is to perform either TEE [39,40] or ICE [41], immediately prior to transcatheter occlusion of ASD. I prefer TEE, so that I can provide undivided attention to the details of device implantation, while an echocardiographer attends to the details of the TEE. Other cardiologists prefer ICE [41] because the procedure can be performed while the patient is awake, although ICE would involve the introduction of a large sheath on the contra lateral femoral vein. ICE or TEE is performed to measure the dimension of the ASD, to define the atrial septal rims, and to image the entrance of all pulmonary veins into the LA. Some TEE examples demonstrating the dimension of the ASD and rims of the ASD are shown in Figure 14, Figure 15, Figure 16, Figure 17, Figure 18 and Figure 19.

After securing information on the diameter of the ASD and excluding any anomalous connections of the pulmonary veins, a thorough assessment of the rims of the atrial septum is undertaken during TEE or ICE. The rims of the ASD are named either on the basis of the location of the septal rim relative to the ASD, such as superior–posterior, superior–anterior, inferior–posterior, or inferior–anterior [30], or the relationship to the adjacent cardiac structures, namely, aortic rim (also called retro-aortic rim), mitral rim, superior vena caval (SVC) rim, and inferior vena caval (IVC) rim, or a combination thereof. Rims smaller than 4 mm are generally considered unsuitable for ASD occlusion with double-disc devices [30,42]. However, this 4 mm limit is not absolute; we and others were successful in accomplishing device closure in patients with septal rims smaller than 4 mm. Nonetheless, evaluation of rims of the ASD is important. Some studies have demonstrated high incidence of deficient septal rims. One such study, by Pondar and associates [43], examining the data on 190 patients, found that the anterior–superior rim was deficient in 42% of patients. They also detected short inferior–posterior rim in 10%, deficient inferior–posterior and posterior rims in 2%, and lacking inferior–anterior, superior–posterior and coronary sinus rims in 1% each [43]. All these rims should be evaluated in multiple echo views, including long-axis, short-axis, four-chamber, and bi-caval projections; some examples are shown in Figure 20. As mentioned in the section on 3D echocardiography, 3D study may be warranted if there is difficulty in accurately defining the septal rims by 2D echo. 

## 7. Balloon Sizing

The technique of balloon sizing the atrial defects was developed by King and his associates in the late 1970s [44]. They withdrew progressively increasing sizes of balloons filled with contrast material across the ASDs to measure their size and found good correlation with those measured at autopsy in animal models and at surgery in live patients. We adopted this technique of dynamic balloon sizing for measuring the size of the ASD on transthoracic echocardiography (Figure 21) prior to the advent of TEE. Once the TEE came into vogue, we used TEE for balloon sizing. 

Because of the cumbersome nature of dynamic balloon sizing and the requirement for the use of large-diameter balloon catheters (requiring large sheaths in the femoral vein), we attempted to estimate balloon size by other techniques of appraisal of the diameter of the ASD, namely, transthoracic echo diameter of the ASD, pulmonary-to-systemic flow ratio (Qp:Qs), and angiographic dimension of the ASD, to evaluate if the balloon-sized diameter of the atrial defect can be forecasted [46]. The statistical analysis in this study revealed that transthoracic echocardiographic ASD diameter had a better correlation (r = 0.82, *p* < 0.001) with balloon-sized atrial defect diameter than with Qp:Qs and angiographic diameter [46]. Based on this correlation, the balloon-sized atrial defect size may be estimated by the following formula: 1.05 × echo + 5.49 mm. In a subsequent investigation [47], we prospectively evaluated this equation in the prediction of the balloon-sized atrial defect diameter by transthoracic echo measures in a separate cohort of 21 subjects. Their ages ranged between 2.5 years and 29 years, with a median of 4.5 years. The predicted balloon-sized atrial defect diameter, derived by the above equation, was 15.7 ± 3.1 mm. This value was not statistically different (*p* > 0.1) from the actual measurement of balloon-sized atrial defect diameter (15.3 ± 4.0 mm). The relationship between the calculated and measured balloon-sized atrial defect diameters was excellent (r = 0.9; *p* < 0.001) (Figure 22). The disparity among the measured and calculated balloon-sized atrial defect diameters was lower than 2 mm in most study subjects. To advance the precision, we recommended that a mean value of the dimensions measured from both long and short-axis projections may be used to estimate the stretched diameter of the ASD. The study conclusions were that the balloon-sized atrial defect diameter can be estimated with precision by two-dimensional precordial echo atrial defect measurements secured before catheter study; this may then be employed to choose the size of the ASD occluding device to be used for the closure of the atrial defect [47].

Subsequently, static balloon sizing of the atrial defects, utilizing either AGA Amplatzer (St. Jude Medical, Inc.) or NuMED PTS (NuMED, Inc., Hopkinton, New York, NY, USA) sizing balloons, was developed and most cardiologists started routinely using this method. After positioning the balloon catheter through the ASD, the balloon is expanded by injecting diluted contrast material till there is no shunting by color Doppler or waisting of the balloon was demonstrated (Figure 23 and Figure 24); over-inflation is avoided. While the balloon is filled, color Doppler imaging of the atrial septum by TEE (or ICE) is performed to detect any additional atrial defects. 

Some studies have indicated that balloon sizing may unintentionally enlarge the ASD size. Because of this reason and the ability to estimate balloon-stretched ASD diameter by TTE size [46,47], as discussed in the preceding portion of this section, the author does not usually carry out balloon sizing of the ASD, but instead relies on the TEE measurements. In addition. the thick rims of the ASD are used to determine the diameter of the atrial defect, without the flail margin (Figure 25)—a technique akin to the method used by Carcagnì and Presbitero [48]. An Amplatzer Septal Occluder (ASO), slightly larger (by 1 to 2 mm) than the dimension of the atrial defect, may be selected for deployment. 

In summary, the author believes that there is no necessity to regularly balloon size the ASDs prior to device occlusion. Estimation of the stretched diameter of ASD by the formula (1.05 × trans-thoracic echo diameter + 5.49 mm) alluded to above [46,47] and measurement of the defect size by TEE utilizing only the thick rims of the ASD, eliminating the thinner and delicate margins (Figure 25), as mentioned above, should suffice. In patients with a substantial difference between the different methods of sizing ASD, balloon sizing is appropriate. Alternatively, 3D echo may be utilized [30,33,34,35] for this purpose. When balloon sizing is performed, TEE (Figure 23 and Figure 25) and ICE are useful in measuring the balloon-sized diameter of the atrial defect.

## 8. Device Occlusion

When the author began performing transcatheter ASD occlusion in the mid-late 1980s, TEE had not yet become a standard method of diagnosis of ASD, nor was it a monitoring tool for ASD occlusion. Fluoroscopy supplemented with transthoracic echo (Figure 21 and Figure 26) was used to guide device occlusion.

Subsequent to the advent of TEE [39,40] and ICE [41], these techniques were employed during device deployment. Transcatheter device occlusion is reasonably similar with all the devices, although some technical differences exist. After securing hemodynamic data, TEE or ICE evaluation of the ASD diameter and septal rims, selective left atrial cineangiography (optional), and balloon sizing (if chosen by the operator), the device delivery sheath is placed in the LA with its tip in the upper-left pulmonary vein. Implantation of the device under TEE/ICE guidance will be described separately for each of the FDA-approved devices. Fluoroscopy is used concurrently as per the operator’s choice. Subsequent to the description of device implantation of each device, issue-related septal rims and multiple defects will be addressed.

### 8.1. Amplatzer Septal Occluder

A suitable-sized device delivery sheath (depending upon the size of the selected ASO) is positioned in the upper-left pulmonary vein. The chosen ASO is fastened onto the delivery wire and the ASO is unscrewed by one rotation (to facilitate eventual device release), and pulled into a loader sheath while immersed in saline. The ASO is inserted into the device delivery sheath while infusing the loader sheath constantly with saline, thus, avoiding unintentional air bubble entry into the delivery sheath. The ASO is pushed forward within the delivery sheath till it arrives at the end of the device delivery sheath, monitored by intermittent fluoroscopy. The delivery sheath is withdrawn from the pulmonary vein so that its tip is in the free LA and the ASO is pushed forward, thus, effecting the release of the left atrial disc. The whole system is slowly pulled back, such that the LA disc is apposed to the left atrial aspect of the atrial septum, closing the ASD; this is performed with continuous echocardiographic (Figure 27A and Figure 28A) and intermittent fluoroscopic monitoring. While holding the device delivery wire stable, the delivery sheath is pulled back, thus, effecting the release of the waist of the device within the ASD. Additional pulling back of the delivery catheter deploys the RA disc in the RA (Figure 27B and Figure 28B). It is important for the echocardiographer to verify that atrial septal margins are sandwiched in-between both the left and right atrial discs (Figure 27B and Figure 28B). Proper safety measures should be taken to circumvent unintentional air entry into the system during the entire process of device loading and implantation.

Following verification of the position of the device by fluoroscopy and TEE (Figure 27B and Figure 28B), color Doppler imaging is performed to detect any outstanding shunt. If the position of the ASO is found to be acceptable (Figure 27B and Figure 28B), the device delivery wire is pulled back and pushed forward gently, known as the Minnesota Wiggle. The satisfactory device location is reconfirmed by TEE (or ICE). Then, the ASO delivery cable is rotated counterclockwise in order to release the device. A repeat TEE (Figure 29) and fluoroscopy are carried out to ensure good device position. If the implanted location of the ASO is unsatisfactory, the device is withdrawn into the delivery sheath and redeployed.

### 8.2. Gore HELEX^®^ Device

The general protocol for transcatheter occlusion with the Gore HELEX^®^ device is largely comparable to that detailed in the ASO section, explained in detail previously [3], and will be briefly described here. The green delivery catheter is positioned in the LA with the aid of a guide wire, which was then taken out. A push-pinch-pull technique is utilized to open up the LA disc. Once the disc is formed, it is withdrawn slowly towards the atrial septum to oppose the LA side of the ASD, while scrutinizing both with fluoroscopy and TEE or ICE (Figure 30A). At this juncture, the green-colored catheter is pulled back over the gray-colored catheter till the mandrel (tan-colored catheter) fits into place with the hub. The green-colored catheter is then fixed in place while the gray-colored catheter is pushed forward to deliver the RA disc on the RA aspect of the ASD (Figure 30B), by utilizing the push-pinch-pull technique. Once the location of the device is verified, TEE or ICE, the device is sealed and then detached (Figure 30C) from the device delivery system. A Minnesota Wiggle type of procedure is not recommended for this device.

### 8.3. Amplatzer Cribriform Device (ACD)

The ACD is useful for occluding fenestrated ASDs (Figure 19, Figure 31 and Figure 32). The method of deployment for the Amplatzer Cribriform Device is akin to that illustrated for ASO in a preceding section. However, the operator must ensure that all the fenestrations of the ASD are covered by the discs of the device (Figure 33), since the connecting waist has no occluding function. Instead, the discs in the device provide the occluding function. It is also important to introduce the delivery sheath through most central regions of the fenestrations; TEE/ICE guidance is helpful in this regard.

### 8.4. GORE^®^ CARDIOFORMASD Occluder (GCO)

The deployment of the GCO device is similar to the Gore HELEX^®^ device described above and is detailed elsewhere [36] for the interested reader. The GCO device is deployed across the ASD via a 10-F delivery sheath under TEE (or ICE) and fluoroscopic control. The device delivery sheath is positioned in the LA with the assistance of a guide wire. Then, the guide wire is withdrawn out of the patient. The LA component of the device is deployed by a simple “push out” technique instead of the more complicated “push-pinch-pull” multiple-step delivery that is required by the Gore HELEX^®^ device. Once the left atrial disk is formed, it is retracted slowly to oppose the LA side of the ASD, while scrutinizing both with fluoroscopy and TEE. At this juncture, the delivery catheter is withdrawn into the lower part of the RA and the RA disc is delivered on the RA side of the ASD, another time by a simple “push out” method. After the location of the device is confirmed by TEE (Figure 34), the GCO is disconnected from the delivery sheath (no Minnesota Wiggle should be performed for this device either). In this example (Figure 34), both the discs of the device are well seated on the atrial septum, although there is a little widening of the upper part of the RA disc (Figure 34B). However, on the three-dimensional reconstruction (Figure 35), the device’s appearance is satisfactory.

### 8.5. Septal Rims

While the waist of ASO serves the occluding function, the outer edges of the discs in the device should straddle the septal rims (Figure 27, Figure 28 and Figure 29). The GCO is a double-disc device and its discs must also be on either side of the septal rim (Figure 34). The echocardiographer should ensure that the margins of the ASD are sandwiched between the discs of the device (Figure 27, Figure 28, Figure 29, Figure 30 and Figure 34).

As reviewed in the section on TEE/ICE, deficient septal rims are problematic. Deficient anterior–superior margin is seen often, especially with large atrial defects [42,43], and indeed, in the author’s personal practice, the majority of patients in whom the author attempted to close atrial defects with different ASD occluding devices were shown to have short anterior–superior rims [49,50,51]. Others had similar experiences [52,53,54]. Most of the defects with deficient anterior–superior rims may be occluded with either established or specially designed techniques [49,50,51,52,53,54]. Nevertheless, atrial defects with an absent or deficient posterior–inferior rim continue to be a difficult task for most interventional cardiologists [42,51,54,55,56]. The role of an echocardiographer is to demonstrate septal rims in multiple views (Figure 14, Figure 15, Figure 16, Figure 17, Figure 18 and Figure 20) and ensure that septal rims are sandwiched between the discs of the device, as illustrated in Figure 27, Figure 28, Figure 29, Figure 30 and Figure 34, prior to disconnecting the device from the delivery cable.

### 8.6. Multiple Defects

The fenestrated atrial septum was addressed in the section on the Cribriform device. Multiple ASDs may be seen in 6 to 7% of patients [57,58,59]. On the basis of the study by Szkutnik and associates [57], we implant a big single device within the larger of the two ASDs, if the distance between the defects is less than 7 mm. If the space between the ASDs is greater than 7 mm, two different occluders are used to close the two defects. When these principles were used, Szkutnik found 100% closure rate at follow-up; the author’s personal experience and that reported by others [58,59] is similar and favorable. Therefore, the TEE (ICE) study should carefully define the sizes of the defects and measure the inter-defect distance. TEE (ICE) monitoring of the device implantation as well as post-device evaluation is similar to that described in the preceding sections.

## 9. Follow-Up after Device Occlusion

Echo-Doppler evaluation is generally recommended on the day following device implantation and at one, six, and twelve months, two years, five years after the procedure, and every five years thereafter. The purpose of such evaluations is to ensure that the device continues to be in stable position, to document improvement in the size of the RV, to monitor ventricular function, to identify any encroachment onto the adjacent structures, to detect residuals shunts, and to scrutinize for any other complications.

### 9.1. Ventricular Dimensions and Function

The RV dimensions generally decrease immediately after ASD closure (*p* < 0.01) (Figure 36) and remain decreased during further follow-up assessment (Figure 36, left panel) [60]. The end-diastolic diameter of the LV does not alter (Figure 36, right panel). Some increase in LV size may be noted and is largely related to and proportionate to the growth of the children. Paradoxical/flat septal motion normalizes during follow-up evaluation (Figure 37).

Global left-ventricular systolic function generally remains unchanged following device closure. Some studies have indicated improvement in heart function after ASD occlusion. Salehian et al. [61] examined myocardial performance index (MPI) in 25 patients at a mean age of 46 years, both before and at a mean of 3 months following device occlusion of atrial defects and found improved MPI values. Both RV MPI (0.35 ± 0.14 vs. 0.28 ± 0.09 {*p* = 0.004}) and LV MPI (0.37 ± 0.12 vs. 0.31 ± 0.11 {*p* = 0.04}) improved at follow-up (Figure 38).

During the normal course of follow-up echocardiographic evaluation, the size and function of the ventricles should be undertaken routinely.

### 9.2. Device Position

Echocardiographic examination is useful in demonstrating the position of the device. In the majority of patients, excellent position of the device immediately following (Figure 29, Figure 30, Figure 33, Figure 34, Figure 35, Figure 39 and Figure 40b), at short-term (Figure 40c and Figure 41) and long-term (Figure 42 and Figure 43) follow-up with all the devices examined [7,36,37,60] can be demonstrated. No residual shunt was seen on color Doppler (Figure 44) in most patients [7,36,37,60]; residual shunts will be reviewed in the next section.

### 9.3. Atrioventricular Valvar Function

In some patients with ASD, tricuspid and/or mitral regurgitation may occur. Both tricuspid and mitral insufficiency resolve in most patients during follow-up echocardiographic evaluation. In an occasional patient, tricuspid or mitral insufficiency may persist and is likely to be related to structural abnormality of the atrioventricular valve. Large devices used for the closure of large-sized ASDs may also cause AV valve disfunction. Both the resolution and persistence of atrioventricular insufficiency may be documented and quantitated by transthoracic echocardiography without difficulty.

### 9.4. Residual Shunts

Residual shunts (Figure 45) are seen is some patients and these residual shunts may be graded, as indicated in Table 1.

With the Amplatzer device, residual shunts have been reported in 4 to 38% at device implantation, which decreased to 1 to 17% at 6- to 12-month follow-up [37]. In our own study [4,63], 16% of patients had trivial-to-small residual shunts at the completion of ASD device implantation. The residual shunts decreased to 11% at six months and to 1.6% at one year after device implantation. However, no shunts were detected fifteen months following occlusion of ASD. With GCO, residual shunts were demonstrated in 23% of patients at device placement, which improved to 14% and 7% at 6- and 36-months follow-up, respectively [36]. The time course for the disappearance of the remaining shunts seen with the buttoned device has been investigated in the past [25,64,65] and one such study is illustrated in Figure 46.

The residual shunts (Figure 45) can easily be documented on transthoracic echocardiographic studies and quantified (Table 1) during follow-up. Centrally located residual shunts are likely to be related to the porosity of the device material and are likely to resolve shortly thereafter. Peripheral residual shunts are likely to be related to insufficient apposition of the device components against the ASD or malposition of the device.

### 9.5. Long-Term Effects

An extensive meta-data analysis [66] revealed preservation of New York Heart Association functional class, improvement in RV dimensions and systolic pressures, and minimal effect on adjusted mortality, but no change in the prevalence of atrial arrhythmias following device closure of ASD. Improvement in RV dimensions and RV/PA systolic pressures can easily be assessed by echo-Doppler studies.

### 9.6. Obstruction of Systemic and Pulmonary Venous Drainage

Large devices used for occluding large ASDs have the potential to interfere with systemic and pulmonary venous drainage, but this is uncommon in the author’s personal experience and that reported in the literature [67,68]; however, such abnormalities can be detected by echo-Doppler studies.

### 9.7. Thrombus Formation

Thrombus formation on and around the device may occur, but is uncommon with the currently used anticoagulation protocols during device implantation and thromboprophylaxis protocols (Aspirin in children and Clopidogrel in adults) in practice during follow-up. Nevertheless, such thrombi may be detected by the currently used echocardiographic studies.

### 9.8. Device Migration

Device migration along with aortic wall perforation by the Amplatzer device has been reported [69,70,71,72]; some of these patients developed fistulae between the aorta and the RA or LA [69,70]. The prevalence of this problem was in 1 in 1000 Amplatzer device implantations [71,72], both in US and international cohorts. This problem was investigated thoroughly [71,72], with the conclusion that this complication is likely to be associated with oversizing of the Amplatzer device (Figure 47), leading to the recommendation that the ASO size must not go above 1.5-times the dimension of the ASD [71,72].

In over 300 Amplatzer Device implantations at my hospital during the past eighteen years, no patient developed fistulae between the aorta and the atria, in spite of careful and systematic follow-up echo-Doppler studies (unpublished observations). We have to presume that this is likely to be connected to avoiding over sizing the device at our hospital. However, relatively low incidence of these complications (1 in 10,000 cases) could also be a factor for not seeing such complication in our 300 patients. Yet, it is important to perform careful echo-Doppler studies periodically during follow-up to detect the aforementioned complications.

## 10. Summary and Conclusions

ASDs are frequent congenital heart defects, both in children and adults. Echocardiography is useful in diagnosing the ASDs, quantifying their size, and defining if indications for ASD occlusion (RV volume overload) are present. Echo studies also are valuable in distinguishing secundum ASDs from other types of atrial defects. An initial selection of patients for closure of ASDs by transcatheter methodology can be made by transthoracic echocardiograms, although the final selection is made by the findings of TEE or ICE and possibly the balloon sizing of ASDs. The procedure of device implantation is largely guided by TEE or ICE along with fluoroscopy. TEE and ICE are specifically useful for assessment of the position of the devices subsequent to their deployment in the catheterization laboratory. Follow-up evaluation with TTE studies is performed to appraise the device position, to identify residual shunts, and to detect rare complications. Based on this review, it may be concluded that the non-invasive transthoracic and invasive transesophageal and intracardiac echocardiographic studies are of immense value, both in the diagnosis of ASDs and in their management. Three-dimensional echo may help with a more accurate definition of complex ASDs.

## Figures and Tables

**Figure 1 diagnostics-12-01494-f001:**
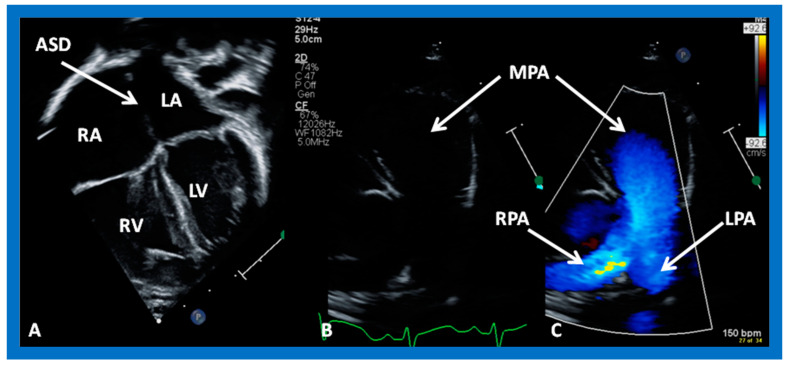
Echo images from apical four chamber (**A**) and precordial short axis (**B**,**C**) projections of a child with an atrial septal defect (ASD) demonstrating dilation of the right atrium (RA), right ventricle (RV) (**A**) and pulmonary arteries (**B**,**C**). LA, left atrium; LV, left ventricle; LPA, left pulmonary artery; MPA, main pulmonary artery; RPA, right pulmonary artery.

**Figure 2 diagnostics-12-01494-f002:**
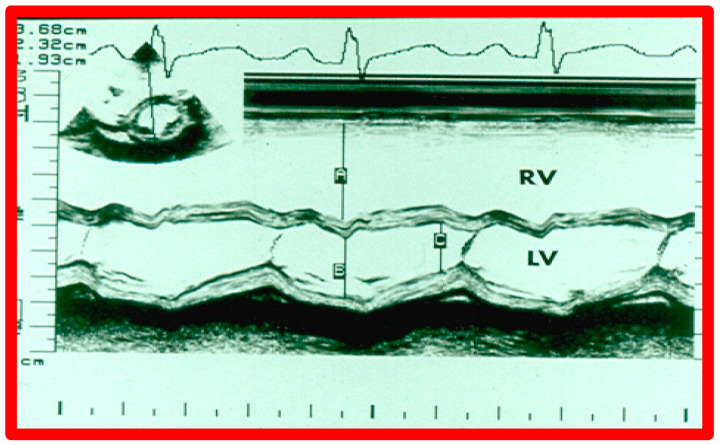
M-mode echocardiogram of a patient with a diagnosis of an atrial septal defect (ASD). It demonstrates dilation of the right ventricle (RV) and paradoxical movement of the interventricular septum in the M-mode recording (2D-derived) in parasternal short axis projection (see lower insert). Normally the ventricular septum functions as a part of the left ventricle (LV); in large ASDs, the septum functions as a part of RV and moves in opposite direction, i.e., paradoxical, also described as “diastolic septal flattening”. The findings are very typical indirect signs of an ASD. A, B, and C stand for measurements of the end-diastolic RV, end-diastolic LV, and end-systolic LV, in that order. The dimensions are shown top left. Modified from Reference [1].

**Figure 3 diagnostics-12-01494-f003:**
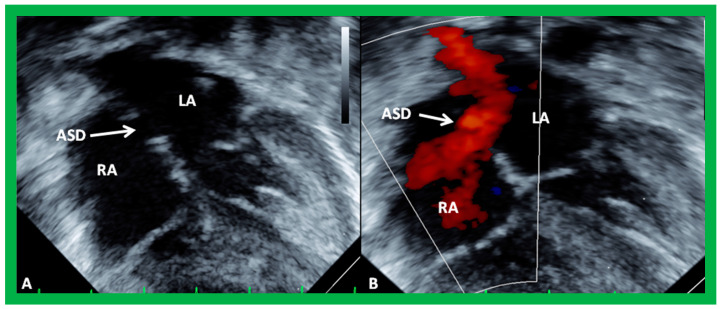
2D echocardiogram (**A**) and color flow imaging (**B**) in subcostal long-axis view, demonstrating an atrial septal defect (ASD) (arrow in (**A**)) with a left-to-right shunt (arrow in (**B**)). Left atrium (LA) and right atrium (RA) are marked. Reproduced from Reference [6].

**Figure 4 diagnostics-12-01494-f004:**
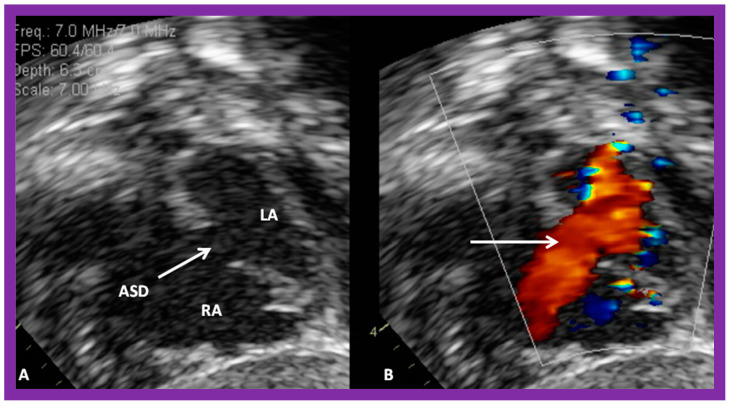
2D echocardiogram (**A**) and color flow imaging (**B**) in subcostal short-axis view, demonstrating an atrial septal defect (ASD) (arrow in (**A**)) with a left-to-right shunt (arrow in (**B**)). Left atrium (LA) and right atrium (RA) are shown. Reproduced from Reference [6].

**Figure 5 diagnostics-12-01494-f005:**
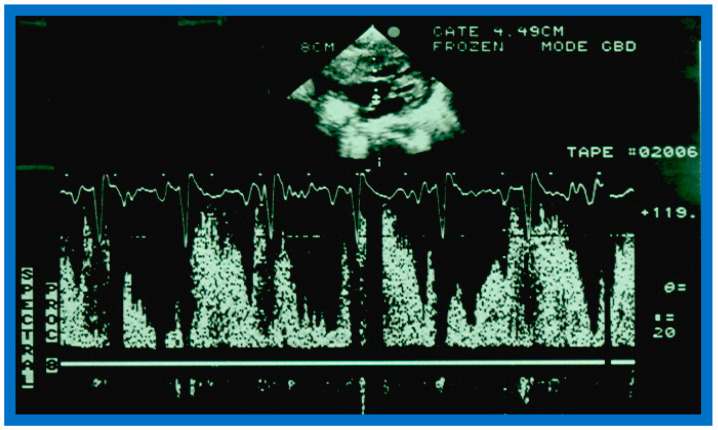
Selected video frame demonstrating flow across an atrial septal defect (ASD) by pulsed Doppler by placing the sample volume on the right side of the ASD (top insert).

**Figure 6 diagnostics-12-01494-f006:**
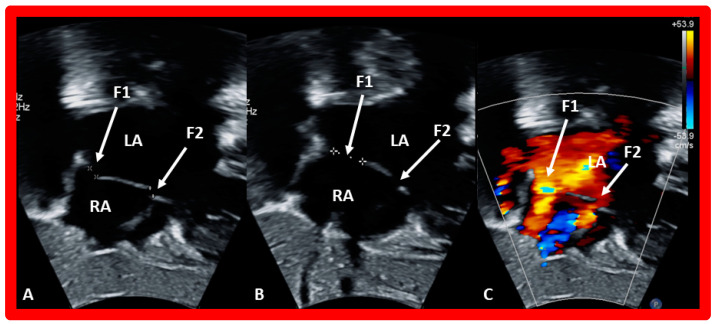
Echo images from subcostal long-axis views of an atrial septum illustrating fenestrated atrial septal defect. Fenestration 1 (F1) (**A**) is much larger and is a somewhat different projection of the atrial septum (**B**). Left-to-right shunts across both fenestrations is shown in (**C**). F2, fenestration 2; LA, left atrium; RA, right atrium.

**Figure 7 diagnostics-12-01494-f007:**
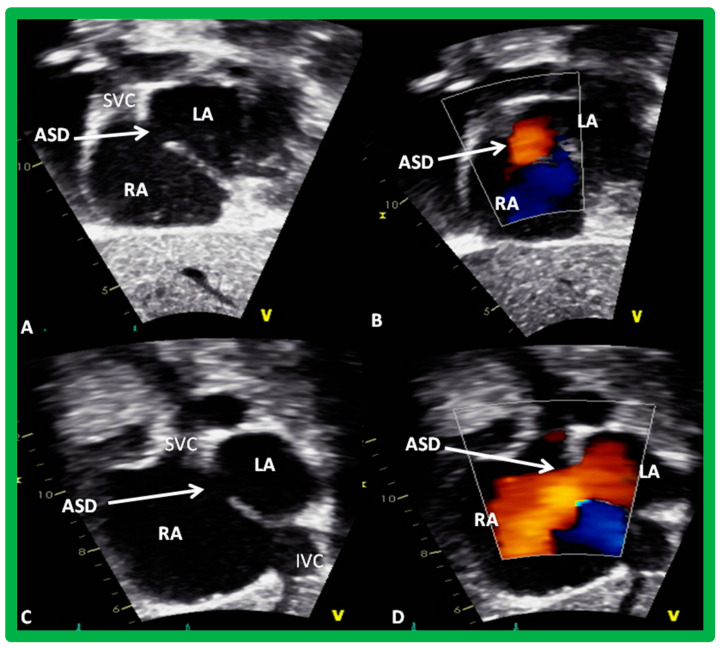
Two-dimensional subcostal long (**A**,**B**) and short (**C**,**D**) -axis projections from a transthoracic echo study illustrating an atrial septal defect (ASD) (**A**,**C**). Color flow imaging (**B**,**D**) demonstrates left-to-right shunt. IVC, Inferior vena cava; LA, left atrium; RA, right atrium; SVC, superior vena cava. Reproduced from Reference [7].

**Figure 8 diagnostics-12-01494-f008:**
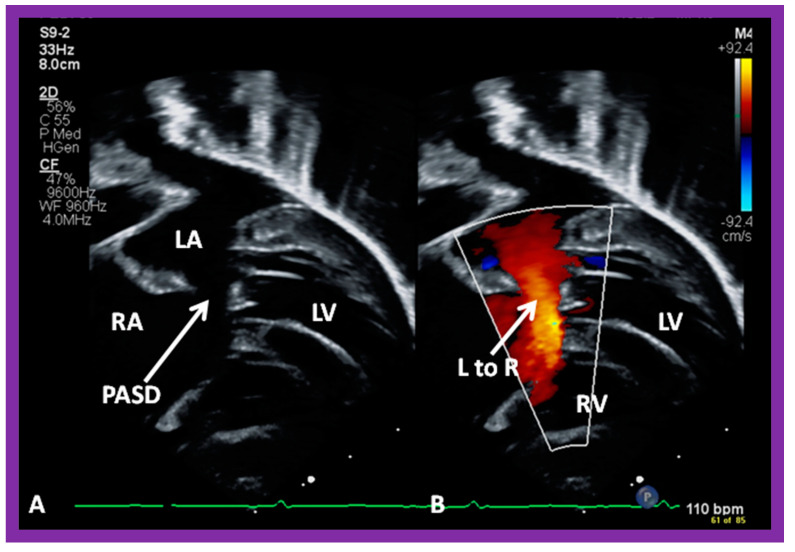
Echo images from modified subcostal four-chambered views illustrating an ostium primum atrial septal defect (PASD) (**A**) with shunting left-to-right (arrow in (**B**)). Left atrium (LA), left ventricle (LV), right atrium (RA), and right ventricle (RV) are labeled.

**Figure 9 diagnostics-12-01494-f009:**
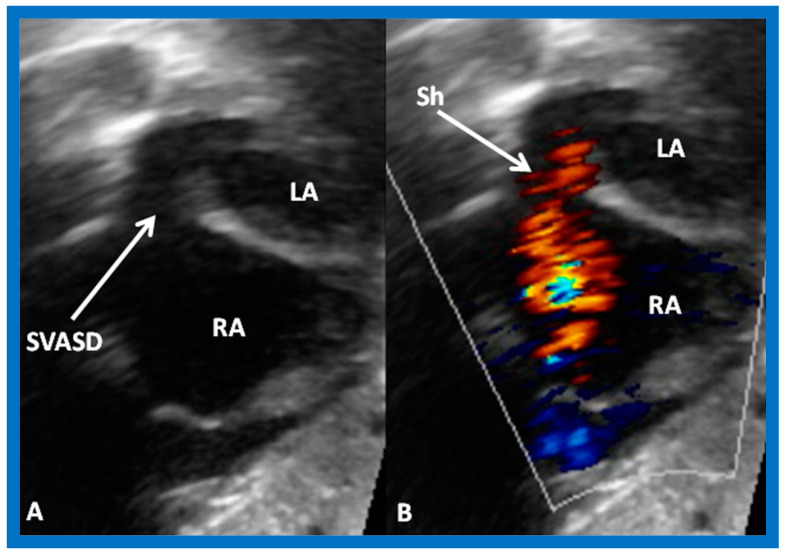
2D (**A**) and color flow (**B**) images from subcostal views demonstrating a sinus venosus atrial septal defect (SVASD) in 2D (**A**) (arrow) with a shunt (Sh) from left to right (arrow in (**B**)). Left atrium (LA) and right atrium (RA) are marked. Modified from Reference [6].

**Figure 10 diagnostics-12-01494-f010:**
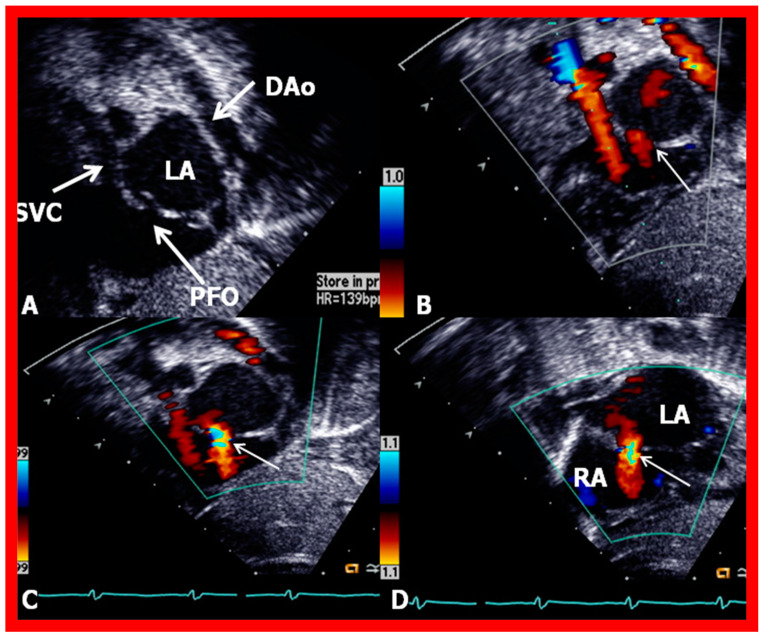
2D (**A**) and color flow (**B**–**D**) images from subcostal short-axis views of the atrial septum, illustrating a patent foramen ovale (PFO) (arrow in (**A**)) with a left-to-right shunt (thin arrows in (**B**–**D**)). Note the overlapping of the septum primum over the septum secundum in (**A**), suggesting that this atrial defect is a PFO. DAo, descending aorta; LA, left atrium; RA, right atrium; SVC, superior vena cava. Reproduced from Reference [6].

**Figure 11 diagnostics-12-01494-f011:**
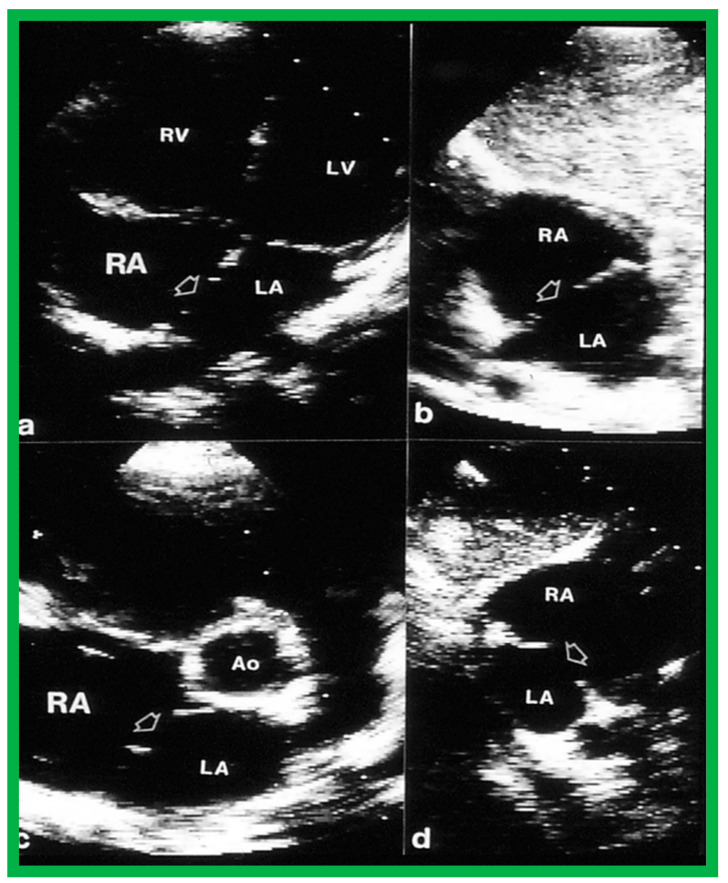
Two-dimensional transthoracic echocardiograms to illustrate the atrial septal rims in multiple views. The apical four-chamber (**a**), subcostal long-axis (**b**), parasternal short-axis (**c**), and subcostal short-axis (**d**) projections are shown. The unfilled arrowheads point to the ASD. Adequate-sized septal rims are seen in each image. This child is deemed to be a suitable candidate for percutaneous device occlusion. Ao, Aorta; LA, left atrium; LV, left ventricle; RA, right atrium; RV, right ventricle. Reproduced from Reference [25].

**Figure 12 diagnostics-12-01494-f012:**
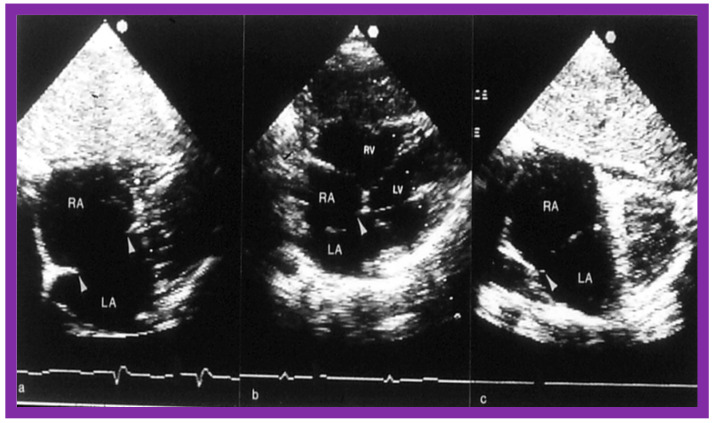
Two-dimensional echocardiograms of atrial septal defects (ASDs) in three different patients to illustrate deficient atrial septal margins. In (**a**) superior and inferior rims are small, in (**b**) the inferior septal rim is deficient, and in (**c**) the superior rim is tiny; the rims are shown with arrowheads. These ASDs are considered unsuitable for transcatheter device occlusion, because the rims of the ASD are inadequate for the device to achieve a good grasp on the atrial septum. LA, left atrium; LV, left ventricle: RA, right atrium; RV, right ventricle. Reproduced from Reference [26].

**Figure 13 diagnostics-12-01494-f013:**
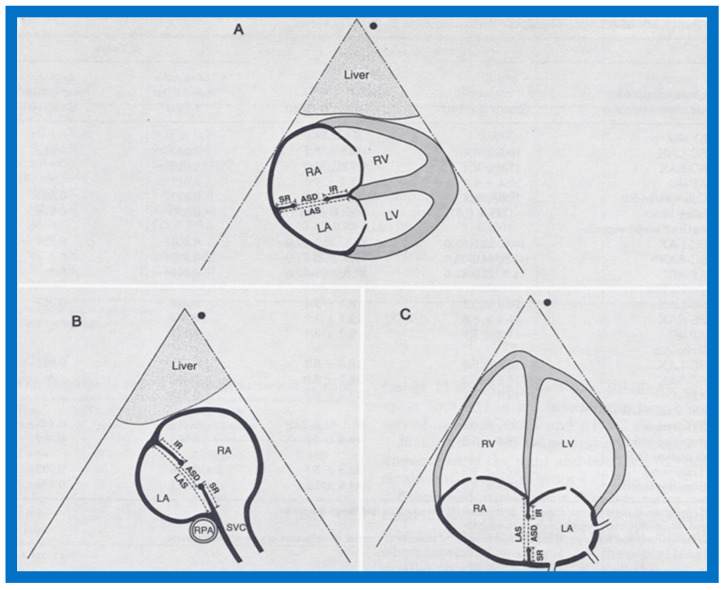
Line drawings of the atrial septum along with the atrial septal defect (ASD) in multiple echo projections, namely, subcostal long axis (**A**), subcostal short axis (**B**), and apical four chamber (**C**), illustrating the methods of measurement of sizes of the defect, length of the atrial septum (LAS), and inferior (IR) and superior (SR) rims of the ASD. Left atrium (LA), left ventricle (LV), right atrium (RA), right pulmonary artery (RPA), and right ventricle (RV), and superior vena cava (SVC) are marked. Replicated from Reference [27].

**Figure 14 diagnostics-12-01494-f014:**
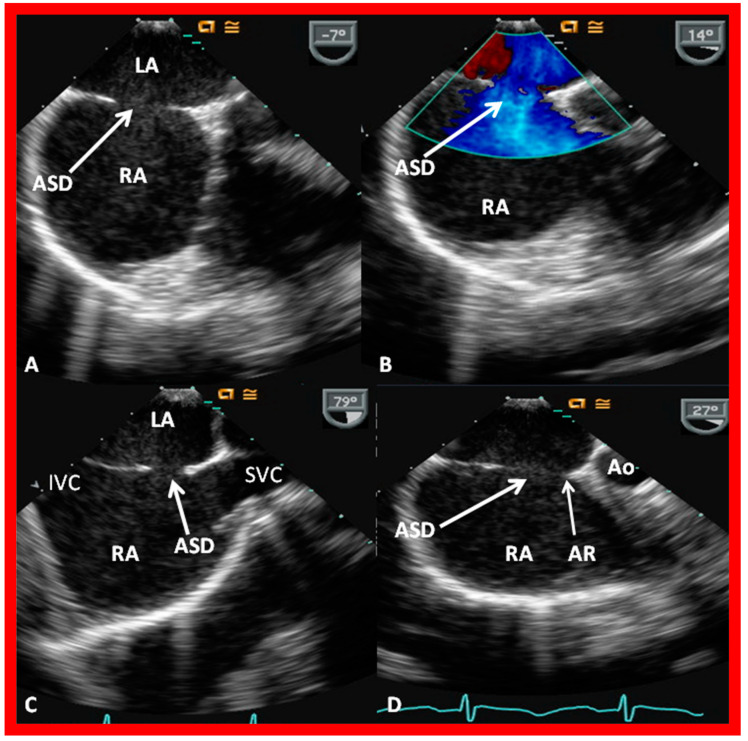
Transesophageal echocardiographic examination demonstrating an atrial septal defect (ASD) in long-axis (**A**,**B**), bi-caval (**C**), and short-axis (**D**) views. Shunt across the ASD is shown by color flow imaging (**B**). Note the very small aortic (anterio-superior) rim (AR) in (**D**). A good-sized superior vena caval (SVC) rim is seen in (**C**). Aorta (Ao), inferior vena cava (IVC), left atrium (LA), and right atrium (RA) are marked. Reproduced from Reference [6].

**Figure 15 diagnostics-12-01494-f015:**
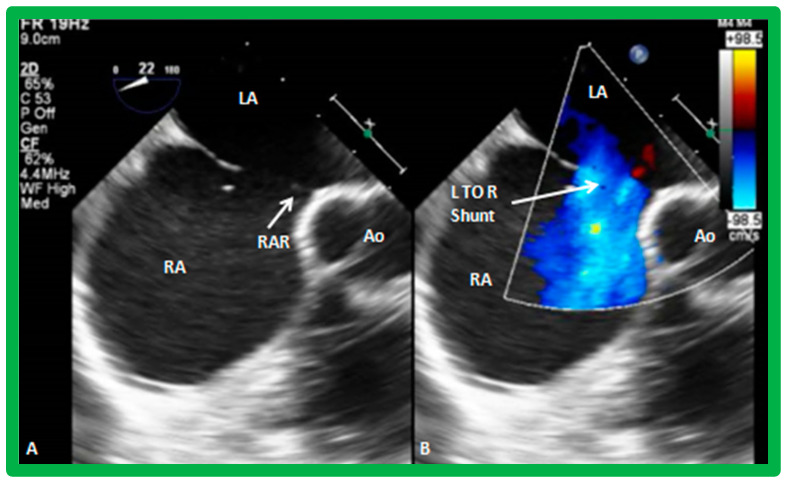
Transesophageal echocardiographic (TEE) images of an atrial septal defect (ASD) in a short-axis projection. An extremely small retro-aortic (RAR) (anterio-superior) rim in (**A**) is seen. Left-to-right (L to R) shunt through the atrial defect by color Doppler is shown in (**B**). Ao, aorta; LA, left atrium; RA, right atrium.

**Figure 16 diagnostics-12-01494-f016:**
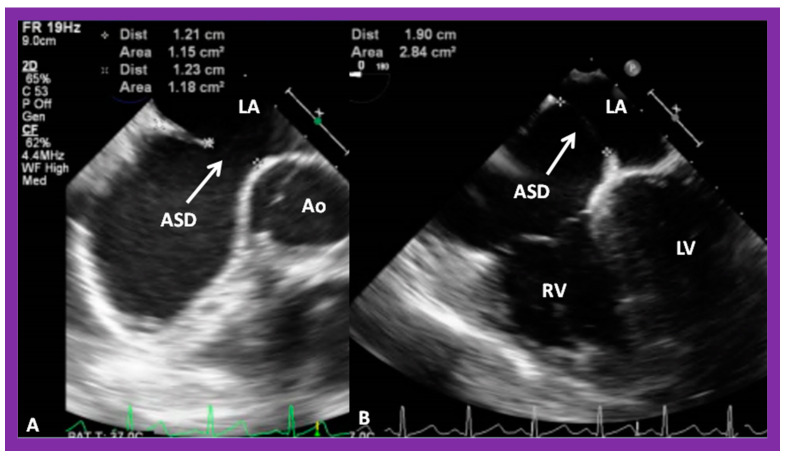
Transesophageal echocardiographic (TEE) study of the atrial septum in short-axis (**A**) and four-chamber (**B**) views demonstrating an atrial septal defect (ASD). Small retro-aortic and mitral rims are seen. Ao, aorta; LA, left atrium; LV, left ventricle; RV, right ventricle.

**Figure 17 diagnostics-12-01494-f017:**
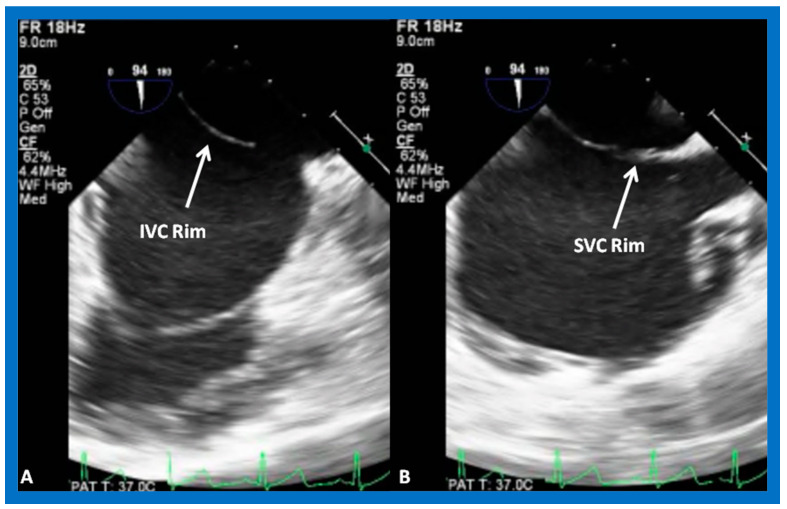
Transesophageal echocardiographic (TEE) study of an atrial septal defect demonstrating adequate-sized inferior vena caval (IVC) (**A**) and superior vena caval (SVC) (**B**) rims.

**Figure 18 diagnostics-12-01494-f018:**
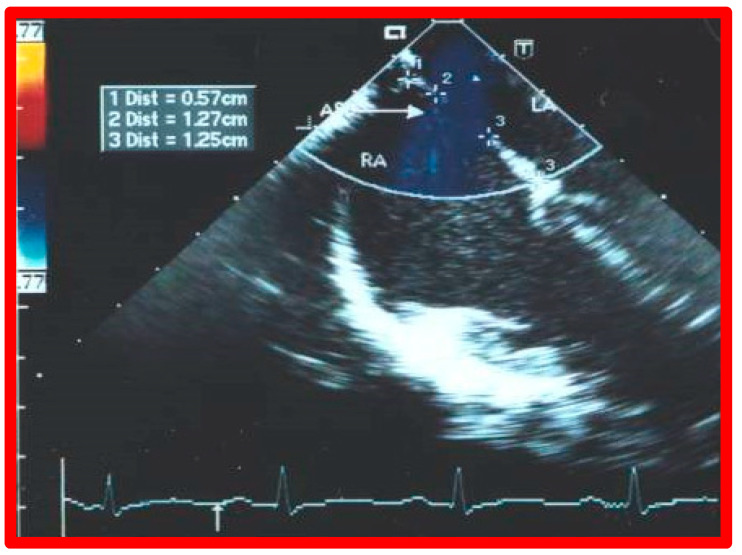
Transesophageal echocardiographic study of an atrial septal defect (ASD) in a four-chamber view demonstrating an ASD (arrow) with shunting left to right. The insert shows dimensions of superior rim (1), ASD (2), and inferior rim (3) in that order. LA, left atrium; RA, right atrium. Reproduced from Reference [3].

**Figure 19 diagnostics-12-01494-f019:**
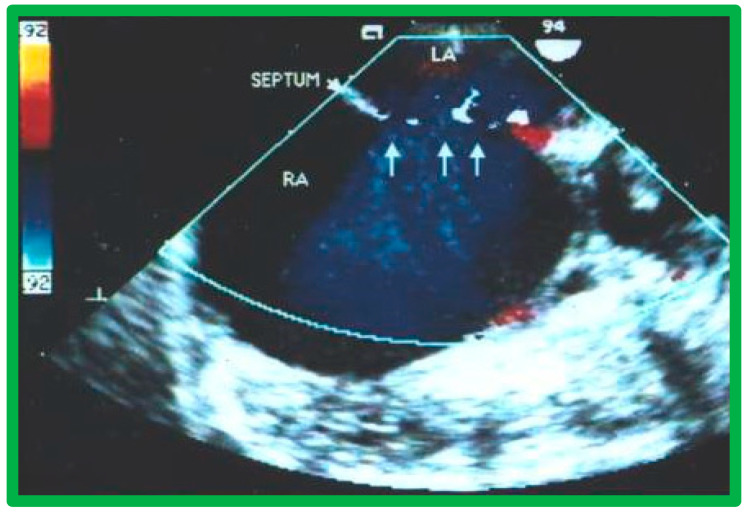
Transesophageal echocardiographic study of a fenestrated atrial septal defect in short projection demonstrating left-to-right shunt across a fenestrated atrial defect (arrows). LA, Left atrium; RA, right atrium. Reproduced from Reference [4].

**Figure 20 diagnostics-12-01494-f020:**
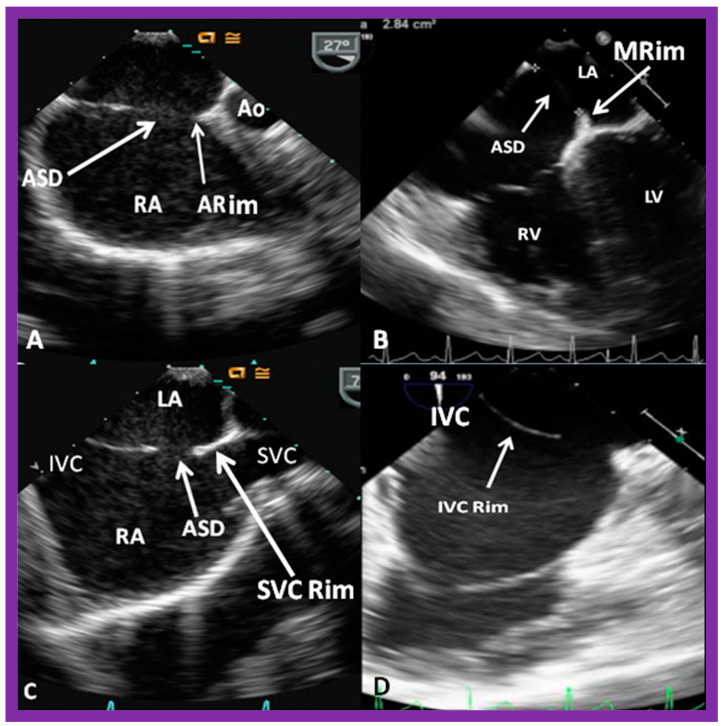
Transesophageal echocardiographic studies of the atrial septum demonstrating aortic (ARim), mitral (MRim), superior vena caval (SVC Rim), and inferior vena caval (IVC Rim) rims in sort-axis (**A**) four-chamber (**B**), and bi-caval (**C**,**D**) views, respectively. Aorta (Ao), atrial septal defect (ASD), inferior vena cava (IVC), left atrium (LA), left ventricle (LV), right atrium (RA), right ventricle (RV), and superior vena cava (SVC) are labeled.

**Figure 21 diagnostics-12-01494-f021:**
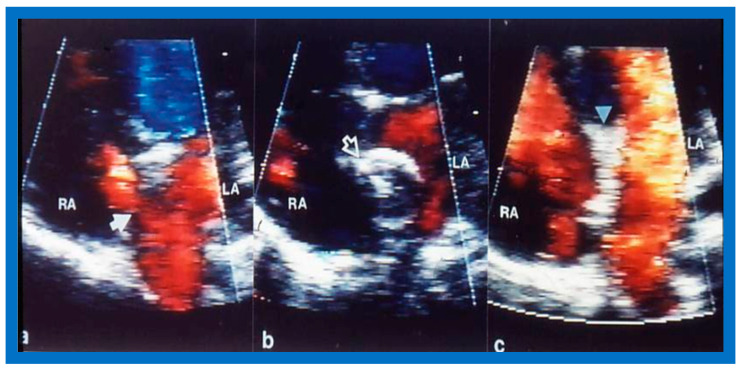
Selected video frames from transthoracic 2D and color Doppler echo demonstrating: (**a**) a shunt across the atrial septal defect (ASD) (filled arrow), (**b**) balloon sizing (unfilled arrow) of the ASD, and (**c**) the device in place (filled arrowhead) without residual shunting. LA, left atrium; RA, right atrium. Reproduced from Reference [45].

**Figure 22 diagnostics-12-01494-f022:**
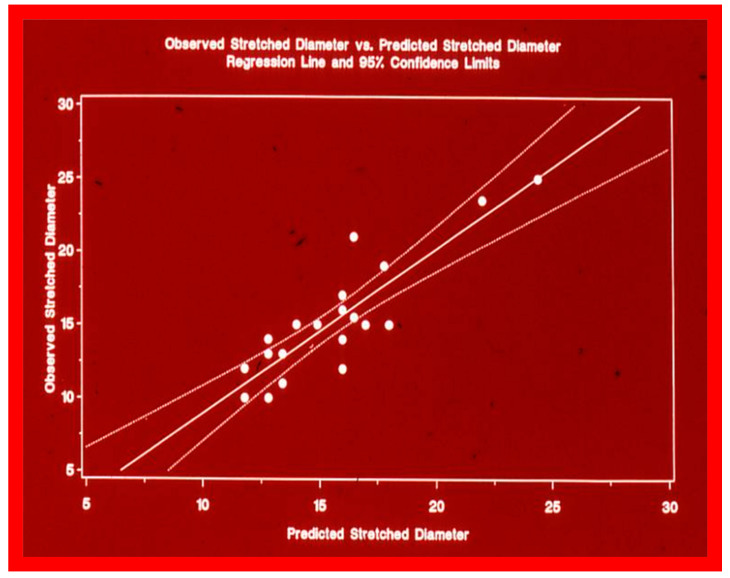
Plot is drawn demonstrating the correlation of the calculated diameter of the atrial septal defect (ASD) by the equation ([1.05 × echo diameter in mm] + 5.49) with the balloon-sized ASD diameter. Statistically important (*p* < 0.001) relationship with an r value of 0.9 was found. Modified from Reference [47].

**Figure 23 diagnostics-12-01494-f023:**
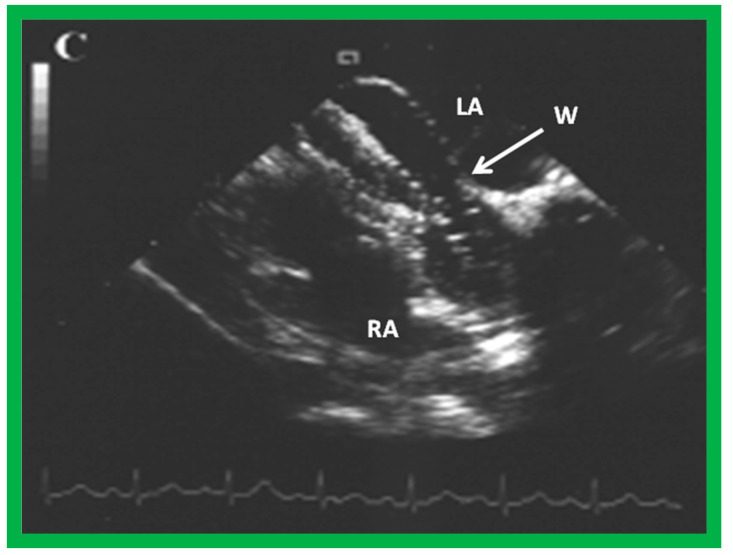
Trans-esophageal echocardiogram demonstrating static balloon sizing of an atrial defect. Waist (W) of the balloon is seen, which measures the size of the ASD. LA, left atrium; RA, right atrium. Reproduced from Reference [4].

**Figure 24 diagnostics-12-01494-f024:**
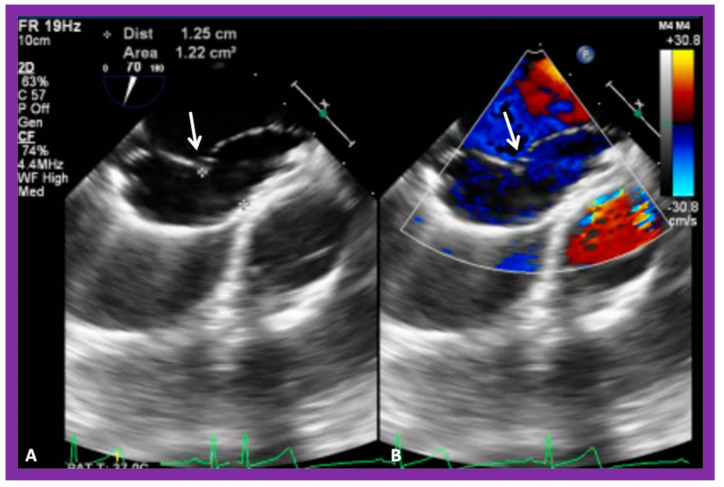
Transesophageal echocardiogram demonstrating static sizing balloon. Note waisting of a sizing balloon (arrow in both (**A**,**B**)). Note that there is no evidence for any shunt in (**B**).

**Figure 25 diagnostics-12-01494-f025:**
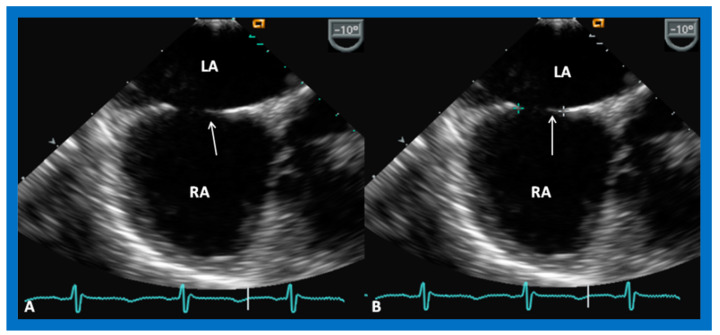
Transesophageal echo of the atrial septal defect, demonstrating the thin margin (arrow) of the atrial septal defect (**A**) which is not included (**B**) in the measurement of the size of the defect. LA, left atrium; RA, right atrium. Reproduced from Reference [6].

**Figure 26 diagnostics-12-01494-f026:**
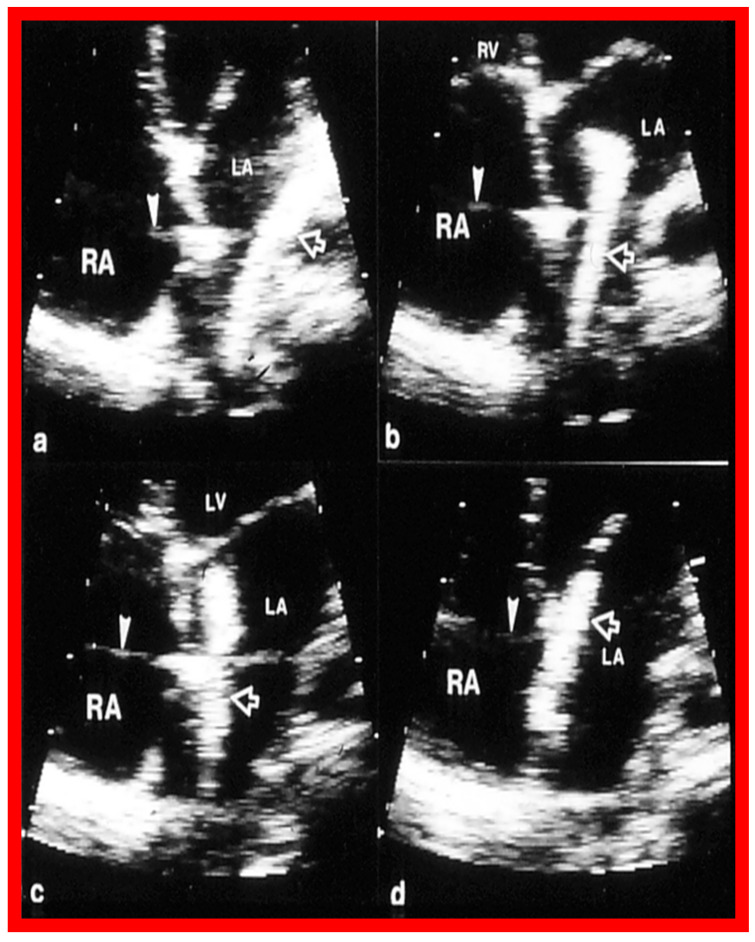
Precordial echo frames from apical four-chamber views of the atria during the deployment of a buttoned device (unfilled arrowheads) to occlude an atrial septal defect (ASD). The occlude component of the device is in the left atrium (LA) and frames (**a**–**d**) demonstrate the occluder being slowly withdrawn towards the ASD to close the atrial defect. The filled arrowheads points to the loading wire attached to the occluder. LV, left ventricle; RA, right atrium; RV, right ventricle. Reproduced from Reference [25].

**Figure 27 diagnostics-12-01494-f027:**
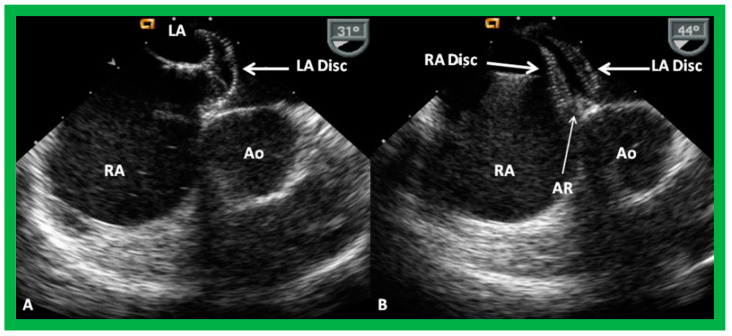
Transesophageal echocardiographic images during the delivery of an Amplatzer Septal Occluder. (**A**). The left atrial disc (LA Disc) (thick arrow) was delivered into the left atrium (LA). (**B**). Both the LA and right atrial (RA) discs (thick arrows) were delivered across the atrial septal defect. It is clearly seen that the aortic rim (AR) (thin arrow in (**B**)) is sandwiched in-between the LA Disc and RA Disc. Ao, aorta. Reproduced from Reference [7].

**Figure 28 diagnostics-12-01494-f028:**
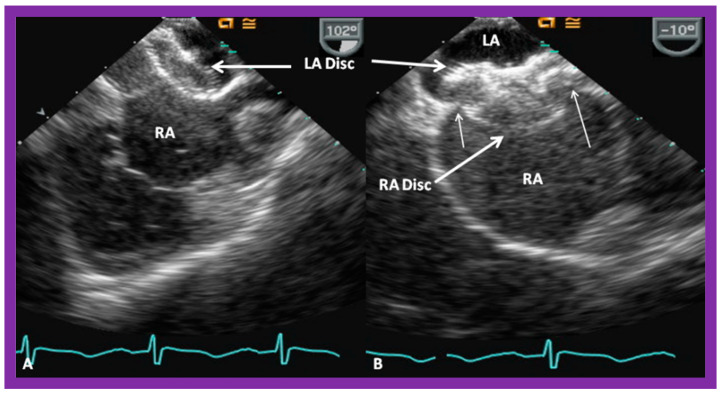
Transesophageal echocardiographic images during the delivery of an Amplatzer Septal Occluder in a different patient. (**A**). The left atrial disc (LA Disc) (thick arrow) was delivered into the left atrium (LA). (**B**). Both the LA and right atrial (RA) discs (thick arrows) were delivered across the atrial septal defect. Again, note the thin arrows pointing to the sandwiching of the septal rims between the LA and RA discs. The labeling notations are those used in Figure 27. Reproduced from Reference [7].

**Figure 29 diagnostics-12-01494-f029:**
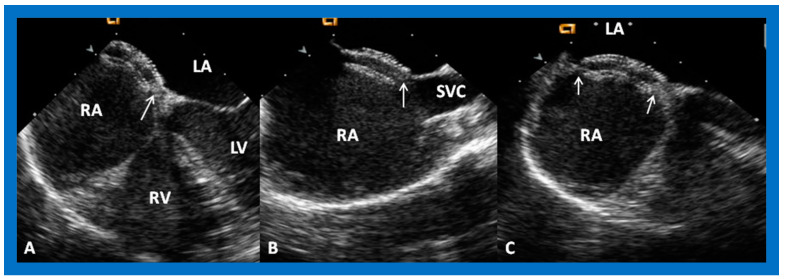
Transesophageal echocardiographic images after the deployment of an Amplatzer Septal Occluder to close an atrial defect, illustrating the location of both the left atrial (LA) and right atrial (RA) discs in four-chamber (**A**), bi-caval (**B**) and long-axis (**C**) projections. It is important to ensure that the septal rims of the atrial defect (thin arrows) are sandwiched in-between the LA and RA discs. LV, left ventricle; RV, right ventricle; SVC, superior vena cava. Replicated from Reference [7].

**Figure 30 diagnostics-12-01494-f030:**
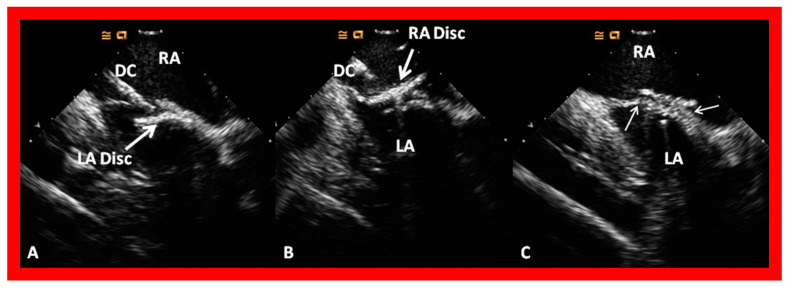
Intracardiac echocardiographic (ICE) images secured during Gore HELEX device deployment to occlude an atrial septal defect (ASD) illustrating the position of the device components. (**A**) The left atrial disc (LA Disc) is in the left atrium (LA). (**B**) The right atrial disc (RA Disc) is in the right atrium. (**C**) The LA and RA discs are on either side of the atrial septum after detachment of device delivery catheter (DC) from the device. Note that the margins of the ASD (thin arrows) are sandwiched in-between LA and RA discs. Replicated from Reference [7].

**Figure 31 diagnostics-12-01494-f031:**
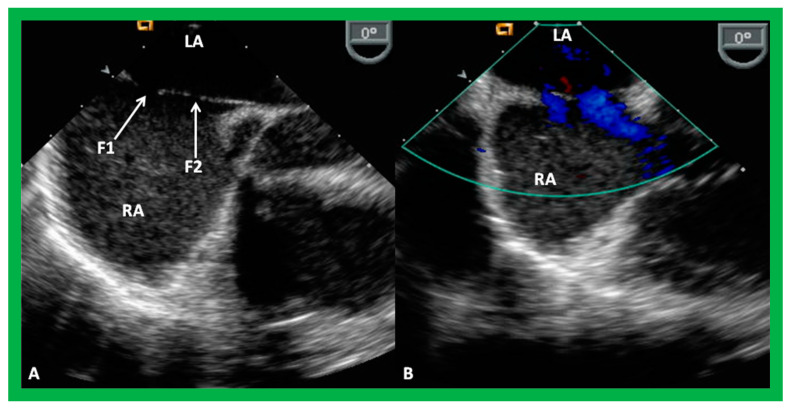
Transesophageal echocardiographic pictures of the fenestrated atrial septal defect by two-dimensional (2D) (**A**) and color Doppler (**B**) imaging. Note the fenestrations (F1 and F2) within the atrial septum in (**A**) and shunting from the left atrium (LA) to the right atrium (RA) in (**B**). Replicated from Reference [7].

**Figure 32 diagnostics-12-01494-f032:**
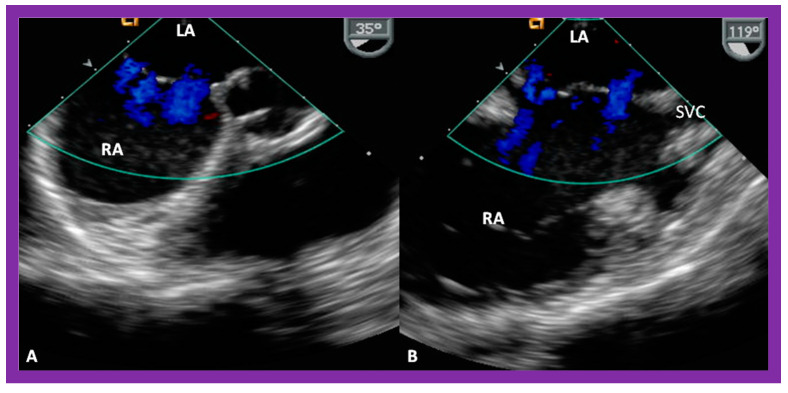
Transesophageal echocardiographic images showing three fenestrations by color Doppler in echocardiographic views (**A**,**B**) different than shown in Figure 31. LA, left atrium; RA, right atrium; SVC, superior vena cava. Reproduced from Reference [7].

**Figure 33 diagnostics-12-01494-f033:**
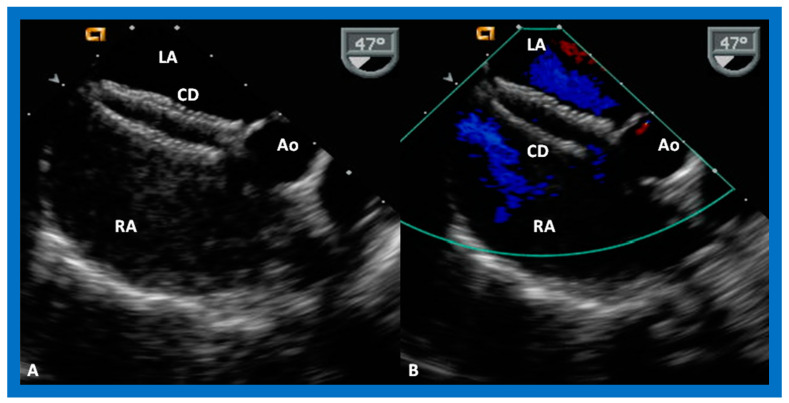
Transesophageal echocardiographic images after the implantation of an Amplatzer Cribriform Device (CD) illustrating the location of right (RA) and left (LA) atrial discs across the atrial septal defect (ASD) (**A**). On color Doppler imaging (**B**), there is no residual shunt (**B**) in a child who had a fenestrated ASD demonstrated in Figure 31 and Figure 32. Ao, aorta; LA, left atrium; RA, right atrium. Reproduced from Reference [6].

**Figure 34 diagnostics-12-01494-f034:**
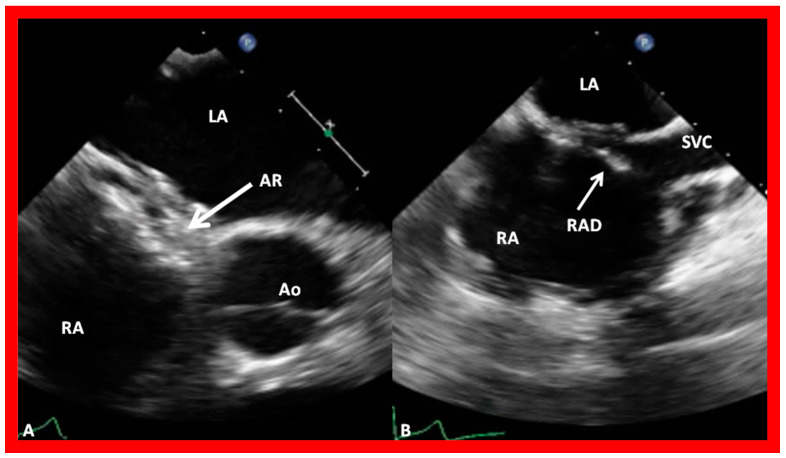
Transesophageal echocardiographic images of the GORE^®^ CARDIOFORM ASD Occluder device after its deployment in two different projections. In the short-axis view (**A**), the aortic rim (AR) (arrow in (**A**)) is visualized sandwiched in-between the right and left atrial discs. In the bi-caval view (**B**) note that there was a little splaying of the upper part of the right atrial disc (RAD). Ao, aorta; LA, left atrium; RA, right atrium; SVC, superior vena cava. Reproduced from reference [37].

**Figure 35 diagnostics-12-01494-f035:**
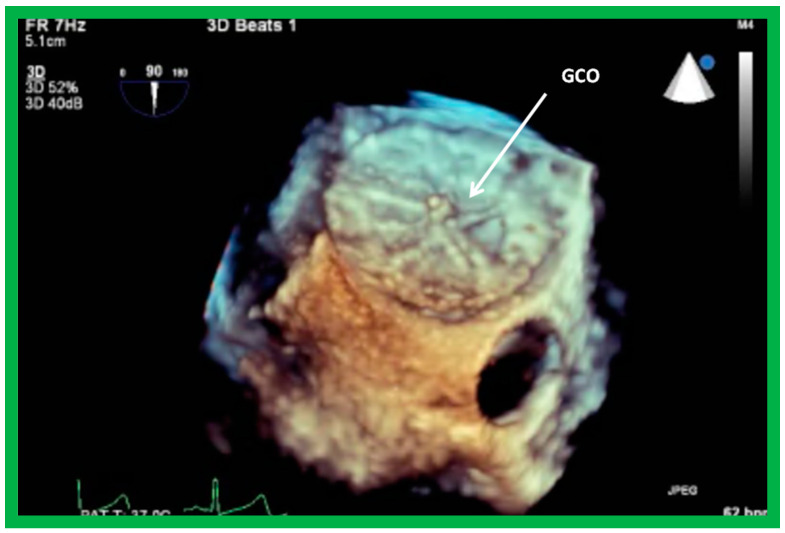
Three-dimensional reconstruction of transesophageal echocardiographic study showing GORE^®^ CARDIOFORM ASD Occluder (GCO) from the right atrial aspect, demonstrating its good position without impinging on the aortic root.

**Figure 36 diagnostics-12-01494-f036:**
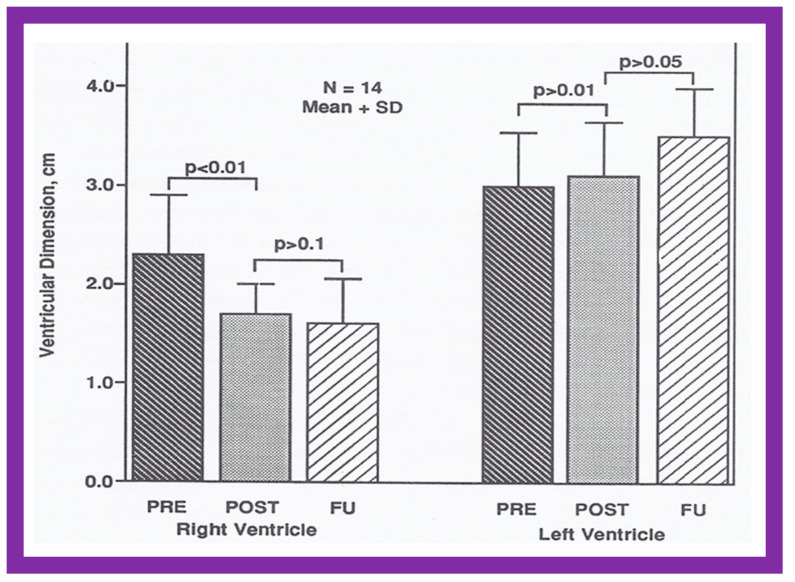
Changes in the end-diastolic diameters of the right ventricle (**left panel**) and left ventricle (**right panel**) following percutaneous occlusion of atrial defects are demonstrated in the above bar graph. The panel on the left illustrates decreased (*p* < 0.01) size of the right ventricle shortly after occlusion (POST) of the atrial septal defect (ASD). At follow-up (FU), there was a further, but not statistically significant (*p* > 0.1), reduction in the end-diastolic measurements of the right ventricle. The panel on the right shows no statistically important alteration (*p* > 05) in the left ventricular dimension either immediately following atrial defect occlusion or during follow-up. The mean + SD (standard deviation) is shown for each measurement. N, number of patients; POST, on the day after ASD occlusion; PRE, before ASD occlusion. Reproduced from Reference [60].

**Figure 37 diagnostics-12-01494-f037:**
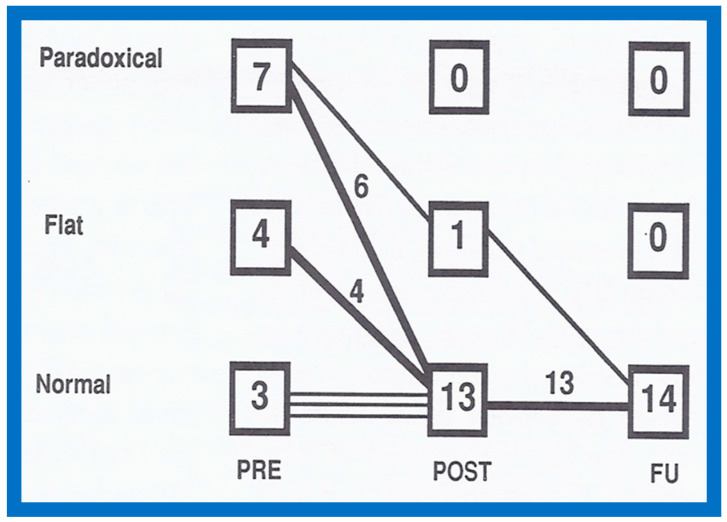
Changes in the inter-ventricular septal motion following percutaneous occlusion of atrial defects are demonstrated in the above graph. Before atrial septal defect (ASD) occlusion (PRE), the inter-ventricular septal motion is either flat or paradoxical in most subjects. Shortly following atrial defect closure (POST), the inter-ventricular septal motion has normalized in all except a single patient. During follow-up (FU), the inter-ventricular septal motion is normal in the entire patient cohort. Reproduced from Reference [60].

**Figure 38 diagnostics-12-01494-f038:**
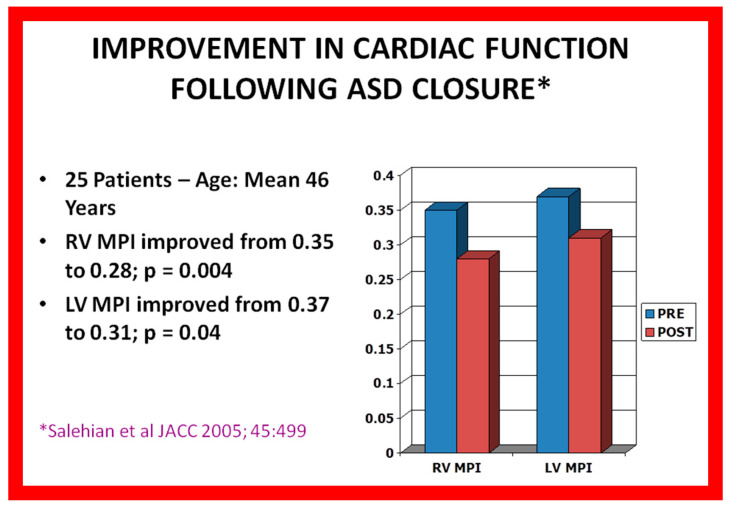
Enhancement of myocardial performance indices (MPI) of right ventricle (RV) and left ventricle (LV) after occlusion of atrial septal defects (ASDs) is illustrated in the above bar diagram. PRE, prior to ASD occlusion; POST, three months following ASD occlusion. Created from the information of Salehian O, et al., JACC, 2005;45:499–504 [61]. Reproduced from Reference [51].

**Figure 39 diagnostics-12-01494-f039:**
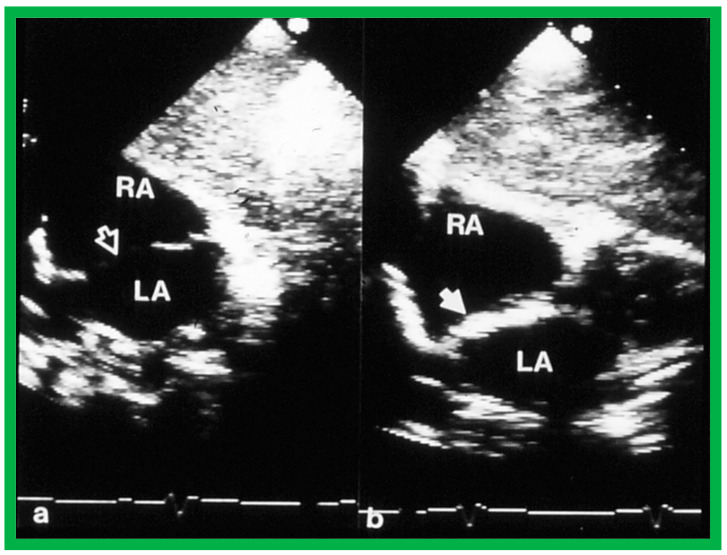
Echocardiographic images from subcostal projections demonstrating an atrial septal defect (ASD) (unfilled arrow) with adequate septal margins before (**a**) and shortly after (**b**) device (filled arrow) deployment. Residual shunt was not seen by color flow Doppler (not shown). LA, left atrium; RA, right atrium. Replicated from Reference [62].

**Figure 40 diagnostics-12-01494-f040:**
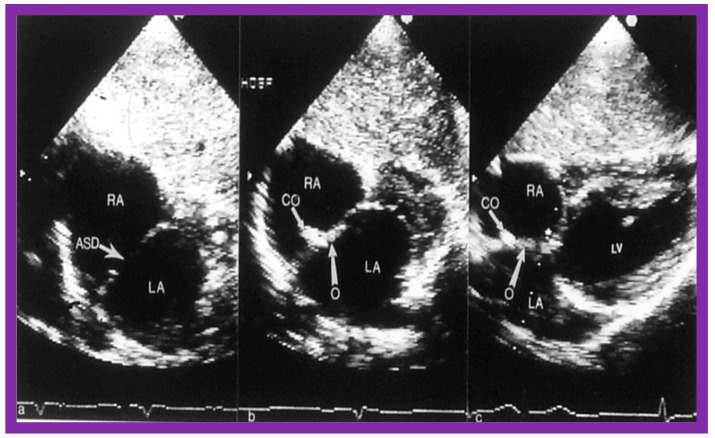
Echocardiographic images from subcostal projections of an atrial septal defect (ASD) (arrow in (**a**)) before occlusion (**a**) with a buttoned device and several hours (**b**) and three months (**c**) following device deployment. The big arrows in both (**b**) and (**c**) identify the occluder component (O) on the left-atrial (LA) side of the atrial septum and the small arrows points out the counter-occluder (Co) on end on the right-atrial (RA) side of the atrial septum. LV, left ventricle. Replicated from Reference [26].

**Figure 41 diagnostics-12-01494-f041:**
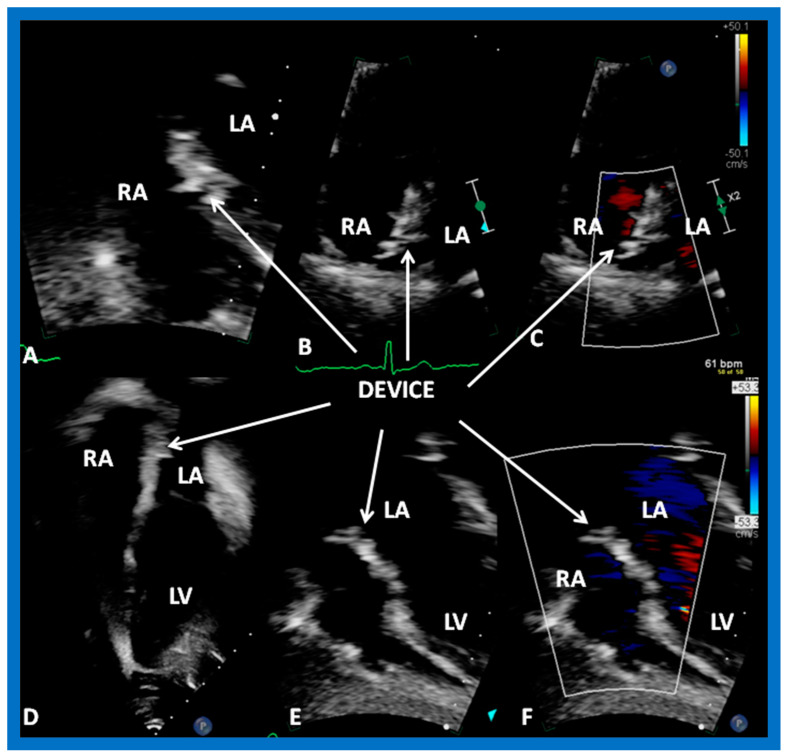
Selected video frames from subcostal (**A**,**E**,**F**) and apical four-chamber (**B**–**D**) transthoracic echo views of the atrial septum, demonstrating GORE^®^ CARDIOFORM ASD Occluder device across the atrial septal defect three months after the device implantation. Good location of the device (DEVICE) without residual left-to-right shunt (**C**,**F**) is shown. LA, left atrium; LV, left ventricle; RA, right atrium.

**Figure 42 diagnostics-12-01494-f042:**
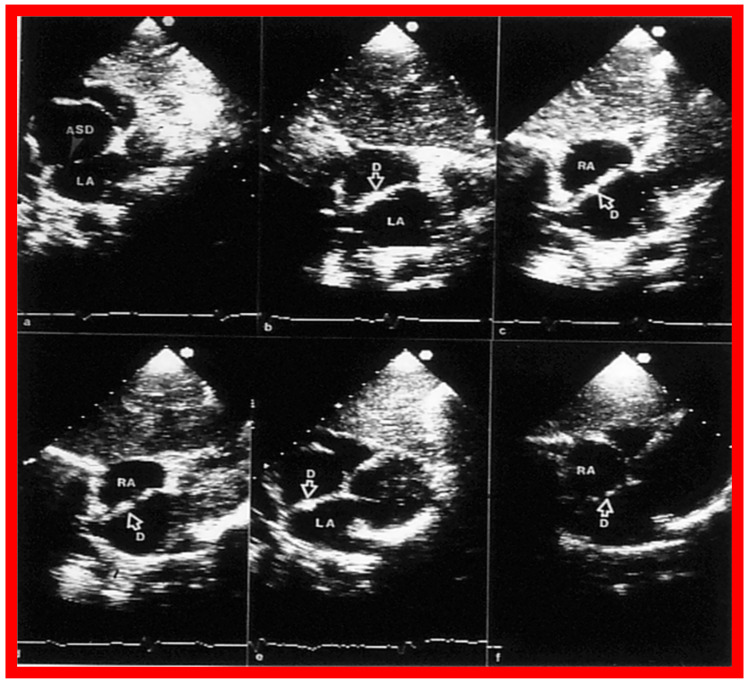
Two-dimensional echocardiographic images in the subcostal projections of the atrial septal defect (ASD) in long-axis views before (**a**), shortly following (**b**), and at one (**c**), six (**d**), twelve (**e**), and twenty-four (**f**) months after ASD occlusion demonstrating the outcome of percutaneous occlusion of an atrial defect with buttoned device. Stable position of the device (D) on the atrial septum (**b**–**e**) is seen. Twenty-four months later (**f**), the device components look to be incorporated into the atrial septal tissue. On pulsed and color Doppler studies simultaneous with 2D examination, no residual shunt was seen (not shown). LA, left atrium; RA, right atrium. Replicated from Reference [60].

**Figure 43 diagnostics-12-01494-f043:**
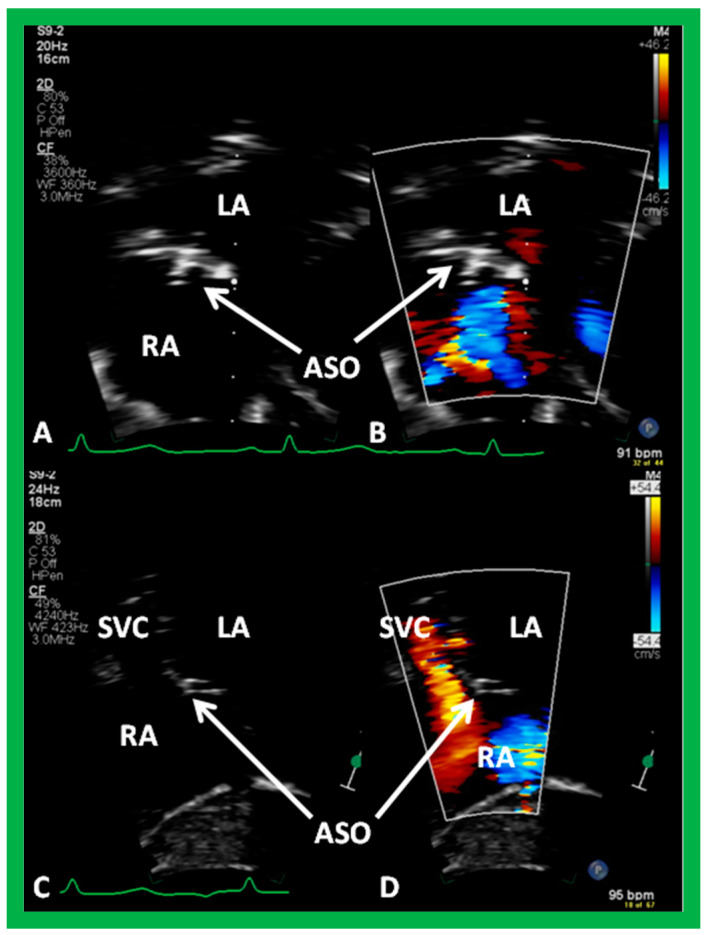
Echocardiographic images from subcostal long (**A**,**B**) and short (**C**,**D**) -axis transthoracic study of the atrial septum, illustrating Amplatzer Septal Occluder (ASO) closing an atrial defect three years after the device implantation; note good position of the ASO device (**A**–**D**) without residual shunt (**B**,**D**). Left atrium (LA), right atrium (RA), and superior vena cava (SVC) are labeled.

**Figure 44 diagnostics-12-01494-f044:**
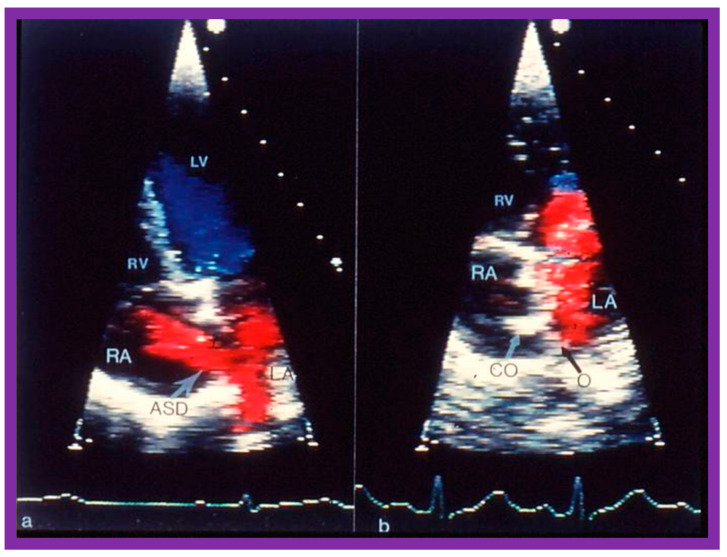
Echocardiographic images from apical four-chamber projections of an atrial septal defect (ASD) illustrating an ASD with a shunt from the left (LA) to the right (RA) atrium before (**a**) and 3 months after (**b**) deployment of a buttoned device. No residual shunt is observed in b. Black arrow in (**b**) shows the occluder (O) on the LA side of the atrial septum while a white arrow displays counter-occluder (CO) end on the RA side of the atrial septum. Left ventricle (LV) and right ventricle (RV) are labeled. Replicated from Reference [26].

**Figure 45 diagnostics-12-01494-f045:**
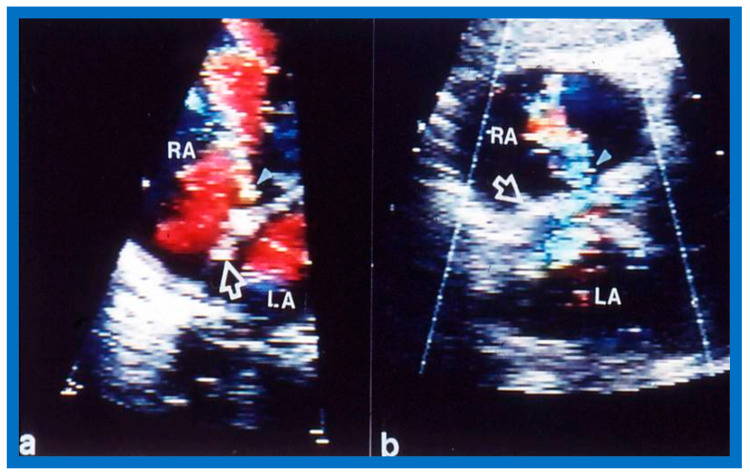
Transthoracic subcostal echocardiographic images of the atrial septum with color Doppler imaging demonstrating the device (unfilled arrow heads) across atrial septal defects in two different patients (**a**,**b**) with residual shunts (filled arrowheads). LA, left atrium; RA, right atrium. Reproduced from Reference [4].

**Figure 46 diagnostics-12-01494-f046:**
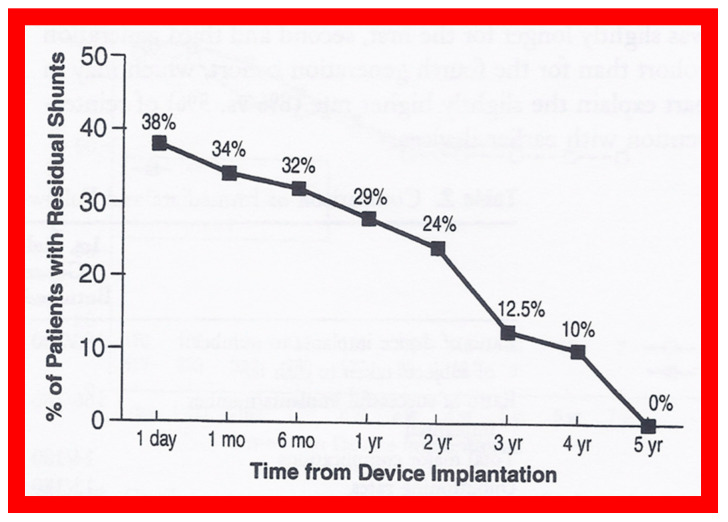
Rates of resolution of residual atrial shunts following occlusion of atrial defects with fourth-generation buttoned devices. The percentage of subjects with residual shunts was calculated as a ratio of the patients with residual shunts divided by the number of subjects examined at that specific follow-up duration. Note the gradual decline in the percentage of patients with residual shunts. mo, month; yr, year. Reproduced from Reference [65].

**Figure 47 diagnostics-12-01494-f047:**
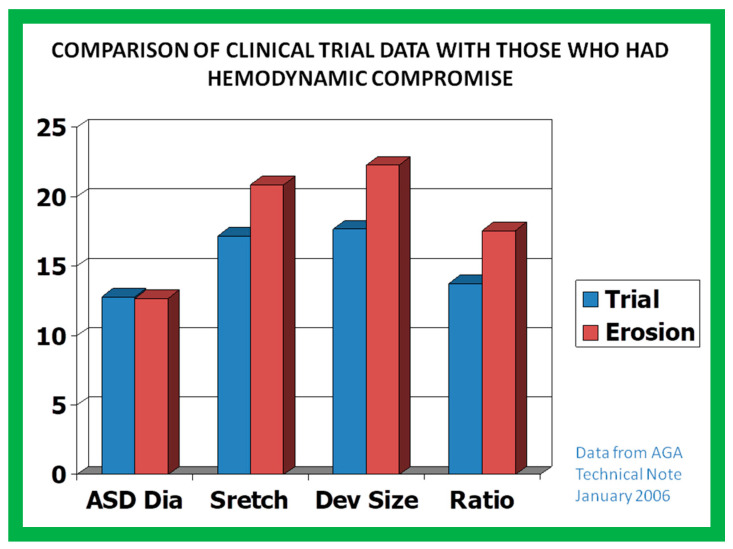
The relation between the diameters of atrial defects and sizes of the Amplatzer device among patients who showed no evidence of perforation (Trial), and those with perforation (Erosion) are shown in this bar diagram. The diameter (Dia) of the atrial septal defect (ASD) was identical in both the described groups (left most panel). However, the balloon-stretched diameter (Stretch), diameter of the device (Dev size), and the ratio of device to atrial defect (Ratio) were larger (2nd, 3rd, and 4th panels) in patients who had perforation than those who did not have perforation (graph was created from the data published in References [71,72]). Based on this information, it was suggested that Amplatzer devices no bigger than 1.5 × the atrial defect diameter should be utilized. Replicated from Reference [51].

**Table 1 diagnostics-12-01494-t001:** Method for classifying left-to-right atrial shunt by echocardiography.

Grade	Criteria
None	No defect by two-dimensional echo No color Doppler disturbance on right atrial side of device No RV volume overload *
Trivial	No defect by two-dimensional echo Minimal color disturbance on right atrial side of device (<l mm width at origin of color Doppler jet) No RV volume overload *
Small	No defect by two-dimensional echo 1 to 2 mm width of color Doppler jet, either in center or on periphery of device No RV volume overload *
Moderate	Defect visualized on two-dimensional echo >2 mm width of color jet RV volume overload * may be present
Large	Defect visualized by two-dimensional echo Large and/or multiple color Doppler jets RV volume overload * is present

RV, Right ventricle. * RV volume overload is defined as an enlarged RV (>95th percentile for patient’s body surface area) and flat to paradoxical inter-ventricular septal motion. Replicated from Reference [25].

## Data Availability

Not applicable.

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
