# Peer review of "Role of Echocardiography in the Diagnosis and Interventional Management of Atrial Septal Defects"

_diagnostics, 2022, doi:10.3390/diagnostics12061494_

Round 1

Reviewer 1 Report

This a clear review on a clinically relevant subject, the manuscript is well written and numerous images are provided. I have a few remarks which it think can slightly improve the paper.

Title: this doesn’t quite cover the scope of the article. Could you change it to e.g. “”role of echocardiography in the Diagnosis and interventional management of Atrial Septal Defects”. Since surgical management of ASDs is hardly mentioned.

Abstract: last sentence, could you change this to: “It is extensively discussed why echo-Doppler is very valuable in diagnosing and managing ASDs”?

Introduction:

“Pulmonary hypertensive disease does not typically occur until late adult life.” Could you provide a reference for this statement?

Patent foramen ovlae should be Patent foramen ovale

Figure 2: could you provide a better quality image? And elaborate on the paradoxal movement of the septum?

Figure 7: could you provide a better quality image?

Page 8. Why is GORE® CARDIOFORM ASD Occluder always underlined?

Page 10 “The disadvantages of 3D echocardiography are low temporal and special resolution and need for offline processing.” Newest software allow to show 3D images while making the images.

Page 16: “The relationship between the calculated and measured stretched ASD sizes was excellent (r = 0.9; p < 0.001) (Figure 27).” When comparing two measurement techniques the relationship should always be great. Bland altman plot or Passing–Bablok regression is a more suitable way of comparing these two measurements.

Figure 26 and 27: can you provide better quality images?

Pages 21 to 23: for a journal that focuses on diagnosis the techniques seem too elaborately discussed. Can you summarize these paragraphs?

Page 22: 8.3 Amplatzer cribiform device: why is this paragraph in bold?

Pages 25 and 26: why are the ventricular dimensions and function so elaborately discussed?

Pages 33 and 34 can you summarize these paragraphs?

Page 35: follow up after surgical closure. Surgical closure is hardly discussed. I think this paragraph can be omitted.

Summary and conclusions: can you add some future perspectives?

Author Response

The reviewer states “This a clear review on a clinically relevant subject, the manuscript is well written and numerous images are provided. I have a few remarks which I think can slightly improve the paper.” Thanks for the complimentary remarks.

The reviewer recommends “Title: this doesn’t quite cover the scope of the article. Could you change it to e.g. “Role of echocardiography in the Diagnosis and interventional management of Atrial Septal Defects”. Since surgical management of ASDs is hardly mentioned.” The title is revised accordingly.

The reviewer suggests “Abstract: last sentence, could you change this to: “It is extensively discussed why echo-Doppler is very valuable in diagnosing and managing ASDs”?” The last sentence of the abstract was changed as per the recommendation of the reviewer.

Introduction:

The author seeks a reference for the statement “Pulmonary hypertensive disease does not typically occur until late adult life.” Could you provide a reference for this statement?” The reference is provided and the sentence is revised also. Subsequent references were renumbered accordingly.

The reviewer identified a typo “Patent foramen ovlae should be Patent foramen ovale”; it is changed accordingly.

The reviewer asks” Figure 2: could you provide a better-quality image? And elaborate on the paradoxal movement of the septum?” Sorry, this from an old publication. Paradoxical was explained in the figure legend.

The reviewer asks “Figure 7: could you provide a better-quality image?” While this is from the author’s old slide collection and does clearly demonstrate the Doppler signal.

The reviewer asks “Page 8. Why is GORE® CARDIOFORM ASD Occluder” always underlined?” The underline is deleted.

The reviewer comments “Page 10 “The disadvantages of 3D echocardiography are low temporal and special resolution and need for offline processing.” Newest software allows to show 3D images while making the images.” Revised accordingly

The reviewer states “Page 16: “The relationship between the calculated and measured stretched ASD sizes was excellent (r = 0.9; p < 0.001) (Figure 27).” When comparing two measurement techniques the relationship should always be great. Bland Altman plot or Passing–Bablok regression is a more suitable way of comparing these two measurements.” This study was done in early 1990s at that time, the statistical consultant, who is a co-authors did not recommend what the reviewer suggests. Too late, it was three decades ago.

The reviewer asks “Figure 26 and 27: can you provide better quality images?” I have carefully examined both these figures; they both look good to me.

The reviewer asks “Pages 21 to 23: for a journal that focuses on diagnosis the techniques seem too elaborately discussed. Can you summarize these paragraphs?” I have gone over this section in attempt to shorten it, but could not make it shorter without loss of comprehensiveness.

The reviewer asks “Page 22: 8.3 Amplatzer cribiform device: why is this paragraph in bold?” In the script that is provided by the Journal, I do not see any bold in this section.

The reviewer asks “Pages 25 and 26: why are the ventricular dimensions and function so elaborately discussed?” It is only 5 lines.

The reviewer asks “Pages 33 and 34 can you summarize these paragraphs?” Shortened and one figure removed.

The reviewer asks :Page 35: follow up after surgical closure. Surgical closure is hardly discussed. I think this paragraph can be omitted.” Deleted.

The reviewer asks “Summary and conclusions: can you add some future perspectives?’ Done.

The author thanks the reviewer for the diligent review and constructive criticism.

Reviewer 2 Report

The author describes his experience with device closure of atrial septal defects, with particular focus on the role of echocardiography in evaluation of the anatomy, suitability for device closure, monitoring during device closure and in follow-up. The review is comprehensive, detailed, and clearly written. It includes many imaging examples.

Specific comments

1)      The author mentions paradoxical (or flat) ventricular septal motion and includes an M-mode example.  I often see this reported as “diastolic septal flattening” and it is assessed from 2D short-axis views. Inclusion of this term would be useful as it is commonly used.  Also, explaining the pathophysiology of this finding would be useful.

2)      Determination of estimated RV or PA systolic pressure is discussed. However, additional review of why this is important would be helpful.  Also, is pulmonary hypertension a contraindication to ASD closure?

3)      I think it would be helpful to describe in more detail which rims are important to measure and what views they should be measured from (i.e. superior and inferior rims from subcostal bi-caval or right sternal border view, retro-aortic and posterior rims from parasternal short-axis, etc . . . .). I find the location of where rims are measured/reported to be variable, and a standardization of this would be helpful. A diagram of the anatomic location of the rims and their corresponding appearance on echo imaging would be quite useful.

4)      When it comes to rim sizes, are there any contraindications to device closure?  Is there a specific rim where it is OK to be deficient or absent, or is there a specific rim where it is not OK? Is the 4 mm cut-off mentioned an absolute? Or in certain situations is it feasible to proceed when < 4 mm.

5)      Regarding atrioventricular valve function, could the author comment on the possibility of a large device distorting AV valve tissue and affecting function, particularly mitral valve?  We make sure the device does not impinge on AV valves during device placement.

6)       Could the comment on the possibility of device obstructing systemic and/or pulmonary venous drainage? How is this assessed by echo?

7)      Could the author comment on the development of thrombus related to the device?

Author Response

The reviewer states "The author describes his experience with device closure of atrial septal defects, with particular focus on the role of echocardiography in evaluation of the anatomy, suitability for device closure, monitoring during device closure and in follow-up. The review is comprehensive, detailed, and clearly written. It includes many imaging examples." Thanks for the complimentary remarks.

Specific comments

1) The reviewer suggests “The author mentions paradoxical (or flat) ventricular septal motion and includes an M-mode example.  I often see this reported as “diastolic septal flattening” and it is assessed from 2D short-axis views. Inclusion of this term would be useful as it is commonly used.  Also, explaining the pathophysiology of this finding would be useful.” The suggestions are incorporated into the figure legend of Figure 2.

2) The reviewer indicates “Determination of estimated RV or PA systolic pressure is discussed. However, additional review of why this is important would be helpful.  Also, is pulmonary hypertension a contraindication to ASD closure?” Agree; a sentence is added as per the reviewer’s suggestion.

3) The reviewer suggests “I think it would be helpful to describe in more detail which rims are important to measure and what views they should be measured from (i.e. superior and inferior rims from subcostal bi-caval or right sternal border view, retro-aortic and posterior rims from parasternal short-axis, etc . . . .). I find the location of where rims are measured/reported to be variable, and a standardization of this would be helpful. A diagram of the anatomic location of the rims and their corresponding appearance on echo imaging would be quite useful.” Septal rims were reviewed in the section. 4. Patient Selection for Device Occlusion,in the  Section on in section 6. TEE & ICE, and the section on 8.5. Septal Rims under Section 8. Device Occlusion

4) The reviewer comments “ When it comes to rim sizes, are there any contraindications to device closure?  Is there a specific rim where it is OK to be deficient or absent, or is there a specific rim where it is not OK? Is the 4 mm cut-off mentioned an absolute? Or in certain situations is it feasible to proceed when < 4 mm.” Four mm is not absolute and I have a sentence in Section 6. TEE & ICE to address reviewer’s suggestions

5) The reviewer comments ”Regarding atrioventricular valve function, could the author comment on the possibility of a large device distorting AV valve tissue and affecting function, particularly mitral valve?  We make sure the device does not impinge on AV valves during device placement.” This was addressed in the section on 9. Follow-up After Device Occlusion, 9.3. Atrioventricular valvar function.

6) The reviewer suggests “Could the comment on the possibility of device obstructing systemic and/or pulmonary venous drainage? How is this assessed by echo?” I have added a section, 9.5. Obstruction of Systemic and Pulmonary Venous Drainage under section, 9. Follow-up After Device Occlusion

7) The reviewer suggests “Could the author comment on the development of thrombus related to the device?” I have added a section, 9.6. Thrombus Formation under section, 9. Follow-up After Device Occlusion

The author thanks the reviewer for the diligent review and constructive criticism.